# Rapid and iterative genome editing in the malaria parasite *Plasmodium knowlesi* provides new tools for *P. vivax* research

Franziska Mohring[1], Melissa Natalie Hart[1], Thomas A Rawlinson[2], Ryan Henrici[1], James A Charleston[1], Ernest Diez Benavente[1], Avnish Patel[1], Joanna Hall[3], Neil Almond[3], Susana Campino[1], Taane G Clark[1], Colin J Sutherland[1], David A Baker[1], Simon J Draper[2], Robert William Moon[1]*

[1]Faculty of Infectious and Tropical Diseases, London School of Hygiene & Tropical Medicine, London, United Kingdom; [2]The Jenner Institute, University of Oxford, Oxford, United Kingdom; [3]Division of Infectious Disease Diagnostics, National Institute for Biological Standards and Control, Health Protection Agency, Hertfordshire, United Kingdom

**Abstract** Tackling relapsing *Plasmodium vivax* and zoonotic *Plasmodium knowlesi* infections is critical to reducing malaria incidence and mortality worldwide. Understanding the biology of these important and related parasites was previously constrained by the lack of robust molecular and genetic approaches. Here, we establish CRISPR-Cas9 genome editing in a culture-adapted *P. knowlesi* strain and define parameters for optimal homology-driven repair. We establish a scalable protocol for the production of repair templates by PCR and demonstrate the flexibility of the system by tagging proteins with distinct cellular localisations. Using iterative rounds of genome-editing we generate a transgenic line expressing *P. vivax* Duffy binding protein (PvDBP), a lead vaccine candidate. We demonstrate that PvDBP plays no role in reticulocyte restriction but can alter the macaque/human host cell tropism of *P. knowlesi*. Critically, antibodies raised against the *P. vivax* antigen potently inhibit proliferation of this strain, providing an invaluable tool to support vaccine development.
DOI: https://doi.org/10.7554/eLife.45829.001

*For correspondence:
rob.moon@lshtm.ac.uk

Competing interests: The authors declare that no competing interests exist.

## Introduction

Malaria remains a serious health burden globally, with over 216 million cases annually (*WHO, 2018*). *Plasmodium falciparum* is responsible for 99% of estimated malaria cases in sub-Saharan Africa. Outside Africa, *P. vivax* is the predominant parasite and causes ~ 7.4 million clinical cases annually. Despite extensive efforts, in 2016 the number of malaria cases were on the rise again for the first time in several years (*WHO, 2018*). Achieving global malaria eradication requires new tools and approaches for addressing emerging drug resistance, relapsing *P. vivax* infections, and emerging zoonotic *P. knowlesi* infections, which represent significant causes of severe disease and death (*Singh and Daneshvar, 2013*; *Hanboonkunupakarn and White, 2016*; *Menard and Dondorp, 2017*).

Although *P. vivax* displays some distinctive features to *P. knowlesi*, including the formation of latent hypnozoites stages in the liver and restriction to reticulocytes in the blood, the two parasites are closely related, occupying a separate simian parasite clade to *P. falciparum* (*Pacheco et al., 2018*). Host cell invasion by *P. vivax* and *P. knowlesi* relies on the Duffy binding proteins (DBP) PvDBP and PkDBPα, respectively, both ligands for human red blood cell (RBC) Duffy antigen/receptor for chemokines (DARC) (*Adams et al., 1990*; *Horuk et al., 1993*; *Singh et al., 2005*;

*Miller et al., 1975*). The critical binding motif of the ligands is the cysteine-rich region 2 (DBP-RII) (*Chitnis and Miller, 1994*), with ~70% identity between PkDBPα and PvDBP (*Ranjan and Chitnis, 1999*). Despite their similarity, PvDBP has also been implicated in both *P. vivax* reticulocyte restriction (*Ovchynnikova et al., 2017*) and as a host tropism factor preventing *P. vivax* from infecting macaques (*Tachibana et al., 2015*). PvDBP-RII is also the leading blood stage vaccine candidate for *P. vivax* (*Ntumngia et al., 2012*; *Payne et al., 2017a*; *Singh et al., 2018*), with antibodies targeting PvDBP-RII blocking parasite invasion in ex vivo *P. vivax* assays (*Russell et al., 2011*). *P. knowlesi* additionally contains two PkDBPα paralogues, namely DBPβ and DBPγ which share high levels of amino acid identity (68–88%) to PkDBPα but bind to distinct receptors via N-glycolylneuraminic acid - a sialic acid found on the surface of macaque RBCs, but absent from human RBCs (*Dankwa et al., 2016*).

Due to the lack of a long-term in vitro culture system for *P. vivax,* vaccine development currently relies on recombinant protein assays, or low throughput ex vivo studies, primate infections or controlled human malaria infections (*Russell et al., 2011*; *Arévalo-Herrera et al., 2005*; *Shakri et al., 2012*; *Payne et al., 2017b*). Thus, higher throughput parasitological assays to assess antisera and antigens, prior to escalation to in vivo work, are desperately needed. The evolutionary similarity between *P. vivax* and *P. knowlesi* means the adaptation of *P. knowlesi* to long-term culture in human RBCs (*Moon et al., 2013*; *Lim et al., 2013*) provides unique opportunities to study DARC-dependent invasion processes in both species. While adaptation of the CRISPR-Cas9 genome editing system to the most prevalent malaria parasite, *P. falciparum* (*Ghorbal et al., 2014*), provided a powerful tool for studying parasite biology, scalable approaches for *P. falciparum* remain constrained by inefficient transfection and very high genome AT-content (averaging 80.6%) (*Gardner et al., 2002*). *P. knowlesi* offers significant experimental advantages over *P. falciparum* including a more balanced genome AT-content of 62.5% and orders-of-magnitude-more-efficient transgenesis (*Moon et al., 2013*; *Grüring et al., 2014*; *Kocken et al., 2002*).

Here, we establish CRISPR-Cas9 genome editing in *P. knowlesi*. Using an optimised and scalable PCR-based approach for generating targeting constructs we define critical parameters determining effective genome editing and apply the technique to introduce epitope/fluorescent protein tags to a variety of proteins with distinct cellular locations. We then use these tools to replace the *P. knowlesi* PkDBPα gene with its PvDBP orthologue, and delete the *P. knowlesi* DBP paralogues to create a transgenic *P. knowlesi* line reliant on the PvDBP protein for invasion of RBCs. The additional deletion of the PkDBP paralogues not only excludes interference through antibody cross-reactivity during growth inhibition assays, but also allows us to demonstrate that, in contrast to previous findings (*Ovchynnikova et al., 2017*), PvDBP plays no role in reticulocyte restriction, but has an effect on macaque/human host cell preference. Finally, we show that antibodies raised against the *P. vivax* antigen are potent inhibitors of *P. knowlesi/*PvDBP transgenic parasites, providing an invaluable tool to support *P. vivax* vaccine development. Thus, we have developed a robust and flexible system for genome editing in an important human malaria parasite and generated essential new tools to accelerate both basic and applied malaria research.

## Results

### Homology mediated CRISPR-Cas9 genome editing is highly efficient in *P. knowlesi*

*Plasmodium* parasites lack a canonical non-homologous end joining pathway, instead relying almost exclusively on homology-directed repair of double-stranded breaks (DSBs), such as those introduced by the Cas9 endonuclease. Effective CRISPR-Cas9 genome editing of malaria parasites therefore requires expression cassettes for the guide RNA and the Cas9 nuclease, and a DSB repair template (donor DNA) containing the desired change, flanked by two regions of homology to the genomic target.

Whilst a variety of approaches have been used in *P. falciparum,* many of the earlier methods embed these elements into two plasmids, each expressing a different drug-selectable marker (*Ghorbal et al., 2014*; *Crawford et al., 2017*; *Mogollon et al., 2016*). This allows for selection of very rare events, but complicates construct design and is not ideal for multiple modifications of a given line – as both selectable markers must then be recycled. As transfection efficiency is

significantly higher in *P. knowlesi* than *P. falciparum* (*Grüring et al., 2014*), we reasoned that we may be able to use a single positive drug selectable marker to cover all the required components for editing. Pairing the guide and Cas9 cassette on a single 'suicide' plasmid (*Lu et al., 2016*) with positive and negative selection cassettes would allow for indirect selection of a separate plasmid containing the repair template, as only parasites that took up the repair template as well as the Cas9 plasmid would be able to repair the DSB. A similar approach to this has been used successfully in *P. falciparum* and has allowed the generation of lines entirely free of resistant cassettes after dilution cloning (*Knuepfer et al., 2017*). Supporting this approach, co-transfection of plasmids expressing eGFP or mCherry revealed that ~30% of *P. knowlesi* transgenic parasites took up both plasmids, although the proportion expressing both declined rapidly in the following days (*Figure 1—figure supplement 1A*). Our two-plasmid CRISPR-Cas9 system comprises one plasmid (pCas9/sg) that provides Cas9, sgRNA (driven by the PkU6 promoter) and a hDHFR-yFCU fusion product for positive/ negative selection, and a second plasmid (pDonor) providing the donor DNA with homology regions (HRs) flanking the DSB for repair by homologous recombination. To test this system, we designed constructs to integrate an eGFP expression cassette into the non-essential *p230p* locus (*Figure 1A* and *Figure 1—figure supplement 1C*). A 20 bp guide sequence targeting a seed sequence upstream of a protospacer adjacent motif (PAM) within the *p230p* gene was cloned into pCas9/sg (pCas9/sg_*p230p*), and a repair template plasmid was synthesized by including an eGFP expression cassette flanked by 400 bp HRs targeting either side of the PAM sequence (pDonor_*p230p*). Both plasmids (each 20 µg) were combined and introduced into *P. knowlesi* schizonts via electroporation (*Moon et al., 2013*) along with control transfections (pCas9/sg without guide sequence and repair template). To simplify synchronisation of parasites, the transfection procedure was altered to additionally include a 2 hr incubation of purified schizonts with 1 µM of the schizont egress inhibitor compound 2, immediately prior to transfection. This compound reversibly inhibits the cGMP-dependent protein kinase (PKG) (*Collins et al., 2013*) and facilitates accumulation of the fully segmented forms required for transfection. Parasites were placed under selection with pyrimethamine for 5 days after transfection and successful integration monitored by PCR. Correct integration at the *p230p* locus was detectable by PCR within 3 days of transfection and only low levels of wild type DNA was detectable after day 11 (*Figure 1B*). Expression of eGFP was confirmed by live microscopy (*Figure 1C*). The eGFP positivity rate was calculated the day after transfection (day 1), to evaluate transfection efficiency (8.4%±2.1 SD). In *P. knowlesi,* expression of GFP can be detected in any parasites that take up the eGFP cassette after transfection, regardless of whether on an episomal or linearised construct. This indicates that 8.4% of parasites successfully take up the construct, but only a very small fraction of these are likely to be integrated at this stage. Only the plasmid pCas9/sg is selected for with pyrimethamine, and so the number of parasites with non-integrated linear donor DNA decreases rapidly over time. The eGFP positivity was assessed again once parasites reached 0.5% parasitemia (day 12), indicating 83.3% (±1.8 SD) of the parasites had integrated the construct (*Figure 1D*); *Figure 1—source data 1*. Parasites transfected with pCas9/sg_*p230p* without providing pDonor_*p230p* were visible in culture several days after the integrated lines. An intact guide and PAM site was detected in these parasites, suggesting that a small population of parasites did not form DSB. Parasites transfected with pCas9/sg without a cloned sgRNA appeared in culture within a few days after transfection, with comparable growth rates to the eGFP plasmid, suggesting the Cas9 expression without a targeting sgRNA is not toxic (*Figure 1E*); *Figure 1—source data 1*. Integrated lines were grown for one week before negative selection with 5-Fluorocytosine and subsequent limiting dilution cloning. Clones were identified using a plaque-based assay (*Figure 1—figure supplement 1B*) previously used for *P. falciparum* (*Thomas et al., 2016*), and 10/10 genotyped clone's harboured correctly integrated, markerless eGFP (*Figure 1F*).

## A three-step PCR method enables rapid, cloning-free generation of donor constructs

*P. knowlesi* readily accepts linearised plasmids for homologous recombination (*Moon et al., 2013*; *Kocken et al., 2002*; *Moon et al., 2016*), so we next tested whether we could use a PCR-based approach for scalable generation of repair templates. As no selectable marker is used within the repair template, this could be easily produced by using PCR to fuse 5' and 3' HRs with the region containing the desired insertion, dispensing with the need for a plasmid back-bone. Modifying a method used for homologous recombination in *P. berghei* (*Ecker et al., 2006*), we developed a

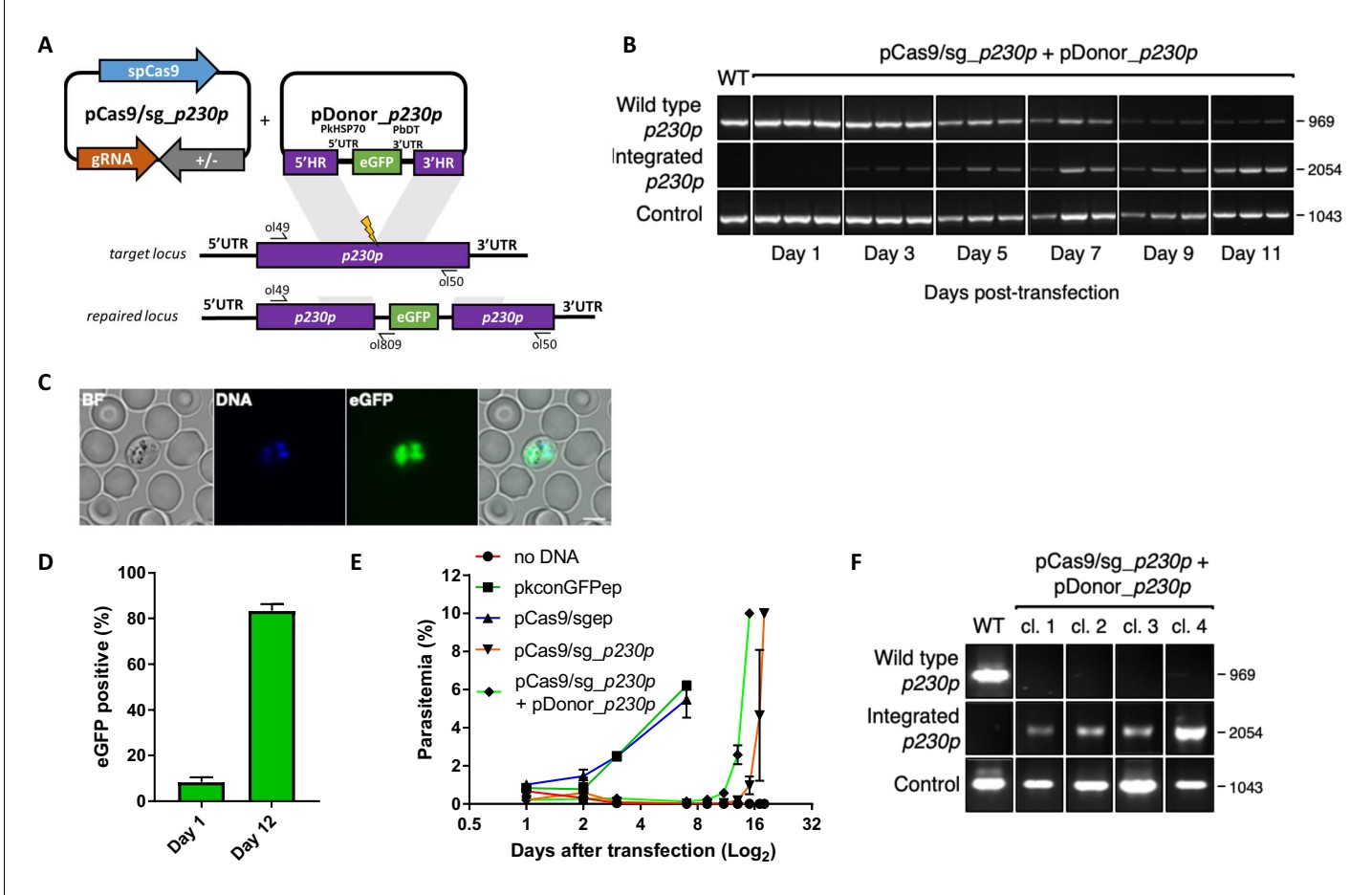

**Figure 1.** CRISPR-Cas9 genome editing in *P.knowlesi*. (**A**) Schematic of CRISPR-Cas9 strategy. Integration of the eGFP expression cassette into the target *p230p* locus via homologous recombination. Arrows indicating oligo positions for diagnostic PCRs. (**B**) Parasites transfected with pCas9/sg_*p230p* and pDonor_*p230p* plasmids were analysed with diagnostic PCRs on consecutive days after transfection. PCR reactions detecting the wild type locus (ol49 +ol50), integration locus (ol01 +ol50) and a control PCR targeting an unrelated locus (ol75 +ol76) using approximately 3 ng/µl genomic DNA. For each day, three transfections are shown. (**C**) Representative live microscopy image of eGFP positive schizont transfected with pCas9/sg_*p230p* and pDonor_*p230p* plasmids. Scale bar represents 5 µm. (**D**) Proportion of eGFP positive parasites (%) counted after transfection with pCas9/sg_*p230p* and pDonor_*p230p* plasmids to show transfection efficiency on day one and integration efficiency after culture reached 0.5% parasitemia (day 12) (n = 3). Error bars denote ±1 SD. (**E**) Graph shows change in parasitemia (%) over time for parasite lines transfected with the dual plasmid Cas9 targeting vectors (pCas9/sg_*p230p* and pDonor_*p230p*), controls without an sgRNA (pCas9/sg), without homology repair template DNA (pCas9/sg_p230p) or with no DNA. A fifth control reaction shows outgrowth of an episomal control plasmid (pkconGFPep) (n = 3). Parasites were placed under drug selection on day 1. Error bars denote ±1 SD (**F**) Parasites transfected with pCas9/sg_*p230p* and pDonor_*p230p* plasmids were cloned by limiting dilution and four clones analysed by diagnostic PCR.

DOI: https://doi.org/10.7554/eLife.45829.002

The following source data and figure supplement are available for figure 1:

**Source data 1.** Source data for graphs.
DOI: https://doi.org/10.7554/eLife.45829.004
**Figure supplement 1.** *P.knowlesi* dual plasmid uptake and plasmid map of pCas9/sg.
DOI: https://doi.org/10.7554/eLife.45829.003

three-step PCR scheme which first amplified the eGFP cassette and 400 bp HRs with eGFP cassette adaptors separately, with the second and third reactions fusing each HR to the eGFP cassette in turn (*Figure 2A*). The addition of nested primers for the second and third PCR step removed background bands and improved robustness. The final PCR construct (HR1-eGFPcassette-HR2) was transfected along with the pCas9/sg_*p230p* plasmid (*Figure 2—figure supplement 1A*), and resultant parasite lines demonstrated integration by PCR (*Figure 2B*), and an eGFP positivity rate of 74% (±8 SD),

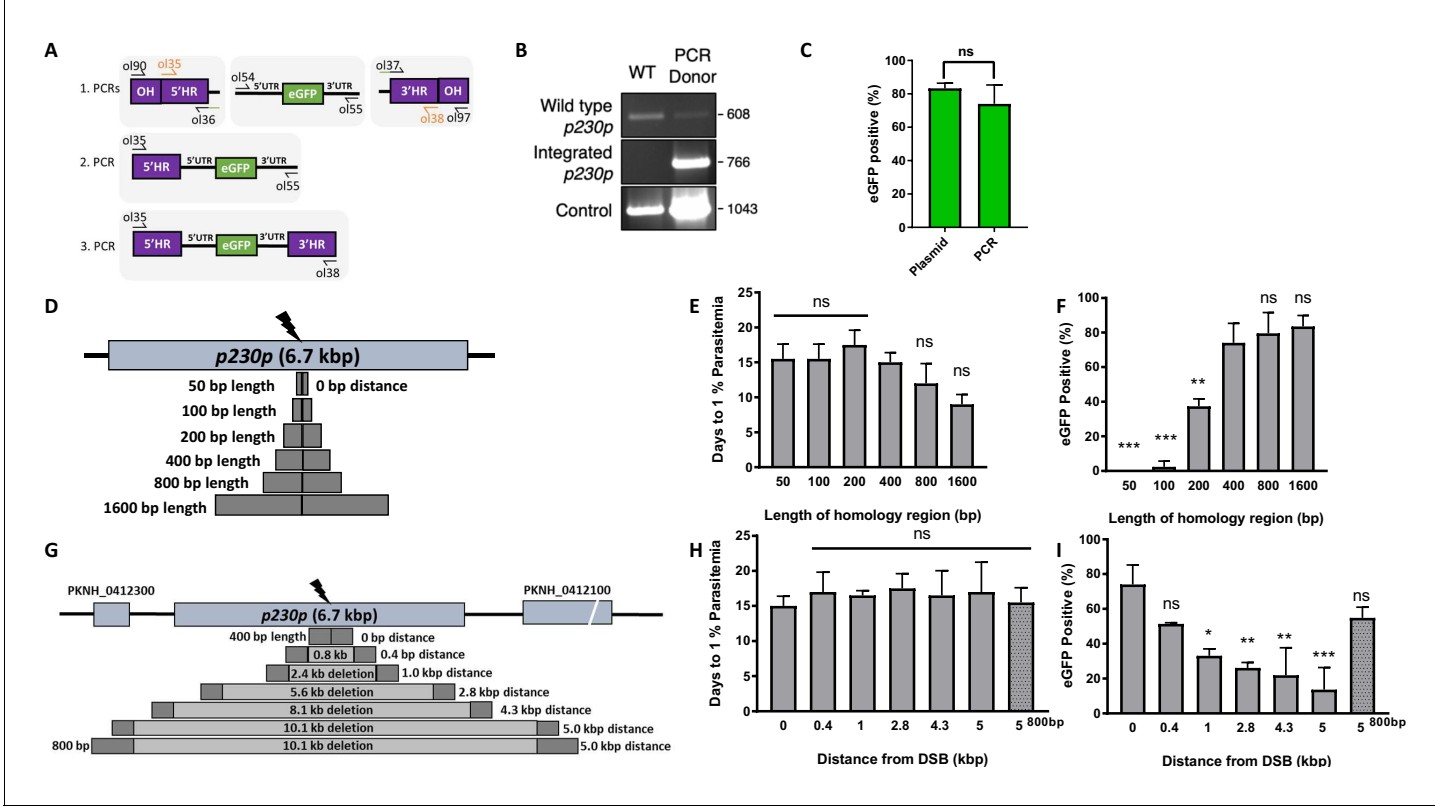

**Figure 2.** Fusion PCR based approach enables cloning-free production of homology repair templates and evaluation of key parameters for efficient homology-driven repair. (**A**) Schematic of the nested PCR method to generate linear donor constructs for transfection. First, homology regions (HRs), with eGFP adaptors in primers ol36 and ol37 and eGFP cassette were amplified by PCR with small overhangs (OH) and gel extracted. In a second nested step 5'HR and eGFP cassette were fused and again in the third step the 5'HR-eGFP product was fused with 3'HR. (**B**) Parasites transfected with pCas9/sg_p230p and PCR repair template (PCR donor), comprised of an eGFP cassette and 400 bp HRs, were analysed with diagnostic PCRs amplifying the wild type p230p locus (ol49 +ol50), integration locus (ol01 +ol50) and a control targeting an unrelated locus (ol75 +ol76). (**C**) After selection for integration, the proportion of eGFP positive parasites (%) was determined by fluorescent microscopy and compared between Cas9 transfections made with 400 bp HR plasmid (pDonor_p230p) or 400 bp HR PCR donor DNA. Data points represent the mean and error bars indicate ±1 SD of two biological independent experiments (n = 2). (**D**) The p230p locus was targeted using PCR donor DNA constructs using HRs with 50–1600 bp length. The bar chart shows, for each of the constructs with HRs of 50 to 1600 bp length, (**E**) the number of days for transfections to reach 1% parasitemia and (**F**) proportion of eGFP positive parasites (%) after selection. All transfections were carried out in two biological independent experiments (n = 2). (**G**) The p230p locus was targeted using PCR donor DNA constructs with HRs placed at varying distance from the Cas9 induced double strand break (DSB). For each construct based on distance to the DSB, the bar chart shows, (**H**) the number of days for transfections to reach 1% parasitemia and (**I**) proportion of eGFP positive parasites (%) after selection. Data points represent the mean and error bars denote ±1 SD of two biological independent experiments (n = 2). Results were all compared to the 400 bp HR construct at 0 kb from DSB as the control using a one-way ANOVA with Dunnett's multiple comparison of means. ns p>0.05, *<0.05, **<0.01, ***<0.001.

DOI: https://doi.org/10.7554/eLife.45829.005

The following source data and figure supplement are available for figure 2:

**Source data 1.** Source data for graphs.
DOI: https://doi.org/10.7554/eLife.45829.007
**Source data 2.** Primer pairs for p230p repair template generation and diagnostic PCRs.
DOI: https://doi.org/10.7554/eLife.45829.008
**Figure supplement 1.** Schematic and genotypic analysis of fusion PCR repair template integration into *p230p* locus.
DOI: https://doi.org/10.7554/eLife.45829.006

similar to that seen for the pDonor_*p230p* plasmid (*Figure 2C*); *Figure 2—source data 1*. The use of a high-fidelity, high-processivity polymerase for the construct production allowed each reaction to be completed in 40–90 min, thus providing a rapid method for generating repair templates.

## Longer HRs increase the integration efficiency and offsets DSB distance efficiency loss

We next used this PCR approach to investigate the optimal parameters and limits of the Cas9 system in *P. knowlesi*. Varying the length of HRs targeting the same *p230p* locus (*Figure 2D*), allowed us to determine the effect on integration efficiency as well as the size limits of the PCR approach. The largest construct generated in this way was 6.1 kb in length (2 × 1.6 kb HRs flanking the 2.9 kb eGFP expression cassette). Attempts to generate a larger 9.3 kb construct (2 × 3.2 kb HRs) failed during the final PCR step. PCR yields were lower for larger constructs, with the 6.1 kb construct yielding half that of the 3.7 kb construct. PCR repair templates with HRs ranging from 50 to 1600 bp generated single specific bands with exception of the 400 bp HRs which contained an additional lower band, due to a primer additionally annealing to a repeat region in HR1 (*Figure 2—figure supplement 1B*). The PCR constructs were transfected together with the pCas9/sg_*p230p* plasmid and integration efficiency monitored. All 6 HR lengths produced evidence of integration by PCR, but the efficiency rapidly declined with HRs shorter than 400 bp (*Figure 2—figure supplement 1D*).

Parasites transfected with 800 and 1600 bp HR constructs were the fastest to reach 1% parasitemia on day 12 and 9 post transfection, respectively (*Figure 2E*); *Figure 2—source data 1*. For the 50 and 100 bp HR constructs no eGFP positive parasites were detected by fluorescence microscopy suggesting very low targeting efficiencies. Constructs with HRs > 400 bp provided GFP positivity ranging from 79% and 81% (*Figure 2F*); *Figure 2—source data 1*, which taken together with PCR yields and transfection recovery time suggest an optimal HR length of at least ~800 bp.

To undertake large gene deletion or replacement experiments, HRs may need to be placed at a distance from the Cas9-induced DSB, and it is well known in other systems that efficiency rapidly declines with distance to DSB (*Byrne et al., 2015*). To determine how distance from DSB affected efficiency of integration, we used the same *p230p* PAM site and moved our 400 bp HRs varying distances away from the DSB, ranging from 0 to 5 kb (*Figure 2G*). PCR repair templates with HRs showed good yields, but again contained an additional lower band for the HRs furthest away from the double strand break (5 kb) (*Figure 2—figure supplement 1C*).

Whilst all transfections were PCR positive for integration and reached 1% parasitemia at similar times (14–20 days) (*Figure 2—figure supplement 1E*, *Figure 2H* and *Figure 2—source data 1*), the integration efficiency declined with distance from DSB. This decline was surprisingly small, with HRs placed even 5 kb away from either side of the DSB yielding a 14% (±18 SD) integration efficiency (*Figure 2I*). Interestingly, we found that extending HR length to 800 bp restored integration efficiencies to 54.8% (±8.7 SD) at a 5 kb distance from DSB (*Figure 2I*; *Figure 2—source data 1*). Thus, HR length can directly offset efficiency losses due to distance from DSB and this system can readily remove genes at least as large as 10 kb in size from a single PAM site, accounting for ~98% of genes in the *P. knowlesi* genome (*Pain et al., 2008*). All primer pairs for template generation and diagnostic PCRs are shown in *Figure 2—source data 2* and primer sequences are listed in *Figure 5—source data 2*.

## Cas9-based PCR constructs enable rapid and flexible gene tagging in *P. knowlesi*

Having demonstrated consistent performance of an sgRNA sequence in the sgRNA/Cas9 suicide vector and PCR constructs for targeting a single control locus, we next sought to determine how robust the system is for targeting a range of loci. We therefore used the PCR-based approach for fusion of fluorescent or epitope tags to proteins of interest (*Figure 3A*). For C-terminal tags, the PCR repair templates were generated by creating fusions of the tag with HRs targeting the 3'end of the gene and the 3'UTR. Similarly, N-terminal tag repair templates were created by flanking the tag with HRs targeting the 5'UTR and 5'end of the coding region. In each case a PAM site was selected so that the 20 bp guide sequence crossed the stop codon (for C-terminal) or start codon (for N-terminal) such that integration of the tag alone, with no other exogenous sequence, was sufficient to disrupt the guide sequence. For genes, such as the Chloroquine Resistance Transporter (CRT), where the PAM site preceded the stop codon, intervening sequences were recodonised when generating the 5'HR to disrupt the PAM site using silent mutations. We selected five genes with disparate subcellular locations and functions to test this approach: the micronemal protein apical membrane antigen 1 (AMA1) (*Bannister et al., 2003*), rhoptry neck protein 2 (RON2) (*Cao et al., 2009*), inner

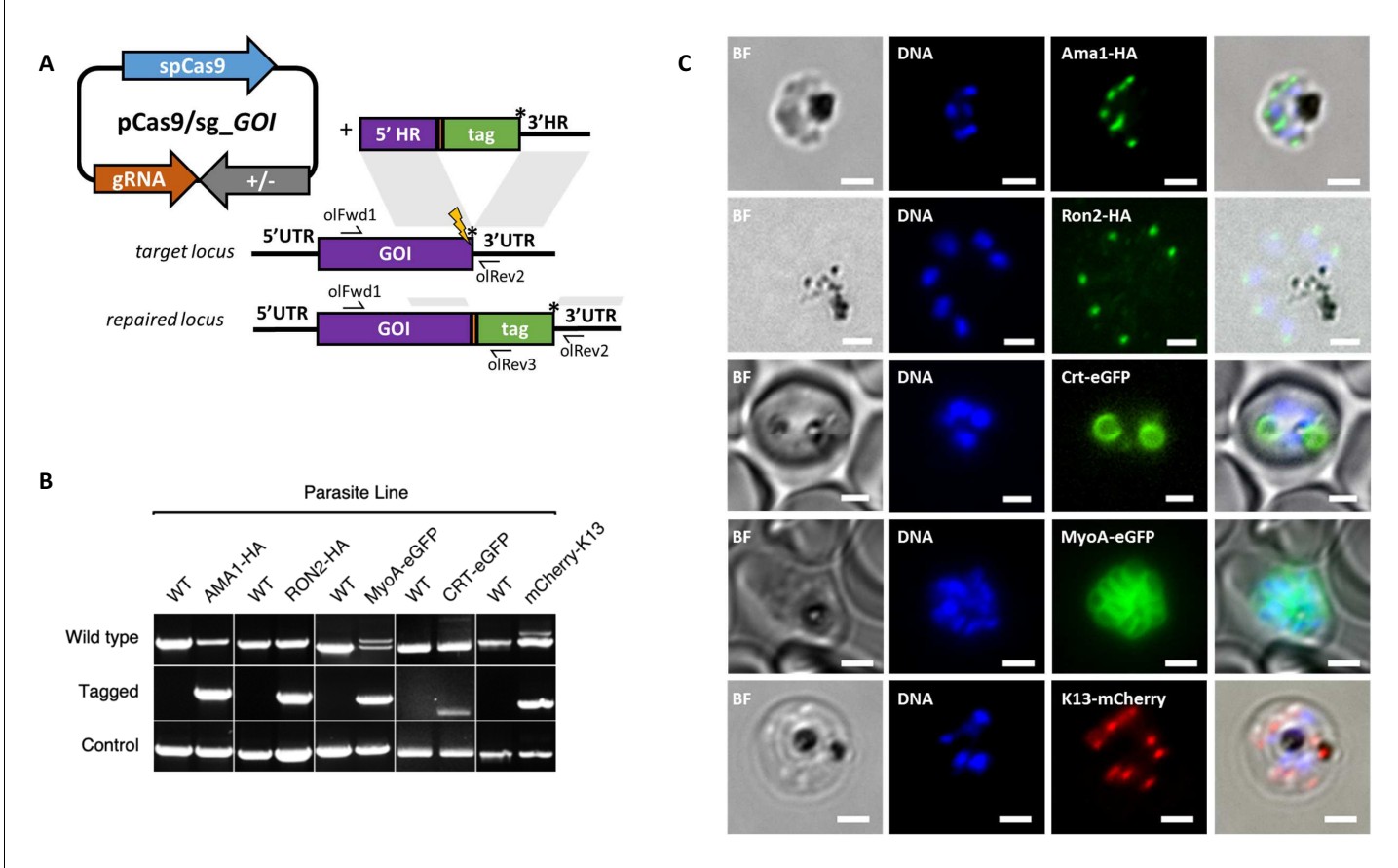

**Figure 3.** CRISPR Cas9 PCR repair templates enable rapid and flexible tagging of parasite proteins. (**A**) Schematic of CRISPR-Cas9 system for C-terminal tagging. pCas9/sg plasmid with gene of interest (GOI) specific sgRNA, is combined with repair template generated by fusion PCR. Lightning bolt indicates Cas9 induced double strand break, which is repaired by insertion of the desired tag. (**B**) Diagnostic PCRs specific to each GOI locus were carried out to amplify the wild type locus (schematic positions olFwd1 +olRev2), integration locus (schematic positions olFwd1 +olRev3) and a control targeting an unrelated locus (ol75 +ol76). Specific primers used for each GOI is shown in *Figure 3—figure supplements 1A–E*, *2*. As no DNA is removed in this process, the wild type specific locus primers also generate slightly larger amplicons in tagged lines, which can be seen as double bands for both the Myosin A and K13 PCRs. (**C**) Representative immunofluorescence images of HA-tagged Apical membrane antigen-1 (AMA1-HA) and Rhoptry neck protein 2 (RON2-HA) parasite lines, and live cell imaging of Chloroquine Resistance Transporter-eGFP (CRT-eGFP), Myosin A-eGFP (MyoA-eGFP) and mCherry-Kelch13 (K13). Panel shows brightfield (BF), DNA stain (blue) and anti-tag antibodies/live fluorescence (green or red) of schizont stage parasites from each line. Scale bars represent 2 μm.

DOI: https://doi.org/10.7554/eLife.45829.009

The following source data and figure supplements are available for figure 3:

**Source data 1.** Primer pairs and guide sequences for generation and analysis of tagged parasite lines.
DOI: https://doi.org/10.7554/eLife.45829.012
**Figure supplement 1.** CRISPR-Cas9 tagging of *P.knowlesi* proteins.
DOI: https://doi.org/10.7554/eLife.45829.010
**Figure supplement 2.** Comparison of *P. knowlesi* and *P. falciparum* 3D7 genes.
DOI: https://doi.org/10.7554/eLife.45829.011

membrane complex protein myosin A (MyoA) (*Baum et al., 2006*), digestive vacuole membrane protein involved in drug resistance CRT (*Ehlgen et al., 2012*), and a protein involved in artemisinin resistance in cytoplasmic foci Kelch13 (K13) (*Birnbaum et al., 2017*). A single sgRNA was selected for each, and repair templates were generated by fusion PCR to incorporate an eGFP, mCherry (both with 24 bp glycine linker) or a hemagglutinin (HA) tag (*Figure 3—figure supplement 1A–E*). An N-terminal tag was used for K13, as previous work in *P. falciparum* suggested that C-terminal tagging affected parasite growth (*Birnbaum et al., 2017*), and C-terminal tags used for all the other targets. All lines grew up quickly after transfection, reaching 1% after between 8 and 15 days, and PCR

analysis indicated that correct integration had occurred (*Figure 3B*). All primers used for generating repair templates and for diagnostic PCRs are shown in *Figure 3—source data 1* and *Figure 5—source data 2*.

Whilst it is, to our knowledge, the first time each of these proteins have been tagged in *P. knowlesi*, all demonstrated localisation patterns were consistent with previous reports for *P. falciparum* (*Figure 3C*). AMA1, MyoA and K13 showed clear bands at the expected size on western blots. The CRT-eGFP fusion protein showed a band at ~50 kDa, in line with work in *P. falciparum* which showed CRT-eGFP migrates faster than its predicted size of 76 kDa (*Figure 3—figure supplement 1F*) (*Ehlgen et al., 2012*). We were unable to visualise a band for RON2-HA most likely due to poor blotting transfer of this 240 kDa protein. Together, these results demonstrate that the fusion PCR approach can be used to tag *P. knowlesi* genes rapidly and robustly at a variety of loci. Analysis of equivalent *P. falciparum* loci revealed only 2/5 had suitably positioned PAM sites, and equivalent UTR regions had an average GC-content of only 11.8% (36% for *P. knowlesi*), suggesting a similar approach would have been more challenging in *P. falciparum* (*Figure 3—figure supplement 2*). All sgRNA sequences with predicted on- and off-target scores that successfully targeted a gene of interest in this study are shown in *Figure 3—source data 1*.

## Transgenic *P. knowlesi* orthologue replacement lines provide surrogates for *P. vivax* vaccine development and DBP tropism studies

Having demonstrated the utility of this technique for rapidly manipulating genes of interest, we next sought to use this system to study *P. vivax* biology. The orthologous RBC ligands PkDBPα and PvDBP, mediate host cell invasion by binding to the DARC receptor on human RBCs in *P. knowlesi* and *P. vivax*, respectively (*Adams et al., 1990*; *Horuk et al., 1993*; *Singh et al., 2005*; *Miller et al., 1975*). PvDBP is currently the lead candidate for a *P. vivax* blood stage vaccine (*Ntumngia et al., 2012*; *Payne et al., 2017a*; *Singh et al., 2018*), thus *P. knowlesi* could provide an ideal surrogate for vaccine testing in the absence of a robust in vitro culture system for *P. vivax*. Whilst likely functionally equivalent, the DBP orthologues are antigenically distinct (~70% amino acid identity in binding region II) so we used genome-editing to generate transgenic *P. knowlesi* parasites in which DARC binding is provided solely by PvDBP. The donor DNA constructs required to fully reconstitute the DBP coding regions were large and UTR and coding sequences for each of the three PkDBP paralogues highly similar at the nucleotide level. Therefore, HRs were amplified from genomic DNA and cloned into a plasmid vector containing the recodonized PkDBPα or PvDBP genes rather than using the 3-step PCR to generate repair templates (*Figure 4—source data 2*). This allowed us to verify amplification of the correct DBP locus and avoid any chance of mutations within the 4.3 kb sized template DNA. We first carried out an orthologue replacement (OR) of the full-length PkDBPα with PvDBP in the *P. knowlesi* A1-H.1 line (PvDBP$^{OR}$) – using a recodonised synthetic PvDBP gene flanked by HRs targeting the 5' and 3'UTRs of the PkDBPα gene (*Figure 4—figure supplement 1A*). Once integrated, this deletes the PkDBPα gene and places the PvDBP gene under control of the PkDBPα regulatory sequences, enabling a precisely matched expression profile. As a control we also exchanged PkDBPα with a recodonised PkDBPα gene (PkDBPα$^{OR}$) using the same sgRNA. Successful integration was readily achieved and limiting dilution cloning resulted in 100% integrated clones for PkDBPα$^{OR}$ and 40% for PvDBP$^{OR}$ (*Figure 4A*). The PkA1-H.1 line relies on the DARC receptor for invasion of human RBCs (*Moon et al., 2013*) and PkDBPα is required to mediate this interaction (*Singh et al., 2005*), thus the successful replacement indicates that the Pv orthologue can complement its role in DARC binding sufficiently well to maintain growth.

*P. knowlesi* contains two DBPα paralogues, DBPβ and DBPγ, which are highly homologous at the nucleotide (91–93% identity) and amino acid (68–88% identity) levels, but are thought to bind to distinct sialic acid-modified receptors unique to macaque RBCs (*Dankwa et al., 2016*). The PkDBPα sgRNA was carefully designed to be distinct to equivalent DBPβ and DBPγ target sequences (85% identical to DBPγ and 47.8% to DBPβ), because, as in other systems, off-target Cas9-induced DSBs are a major issue (*Figure 4—figure supplement 1D*) (*Zischewski et al., 2017*; *Wagner et al., 2014*). We therefore sequenced the four most similar target sequences, including one in DBPγ, in the PvDBP$^{OR}$ lines (*Figure 4—figure supplement 2*) and did not detect any off-target mutations, suggesting that as for other malaria parasites (*Ghorbal et al., 2014*) the absence of non-homologous end joining ameliorates the potential for off-target mutations. However, diagnostic PCRs for DBPβ failed, as well as PCRs in genes flanking the DBPβ locus. Whole genome sequencing and

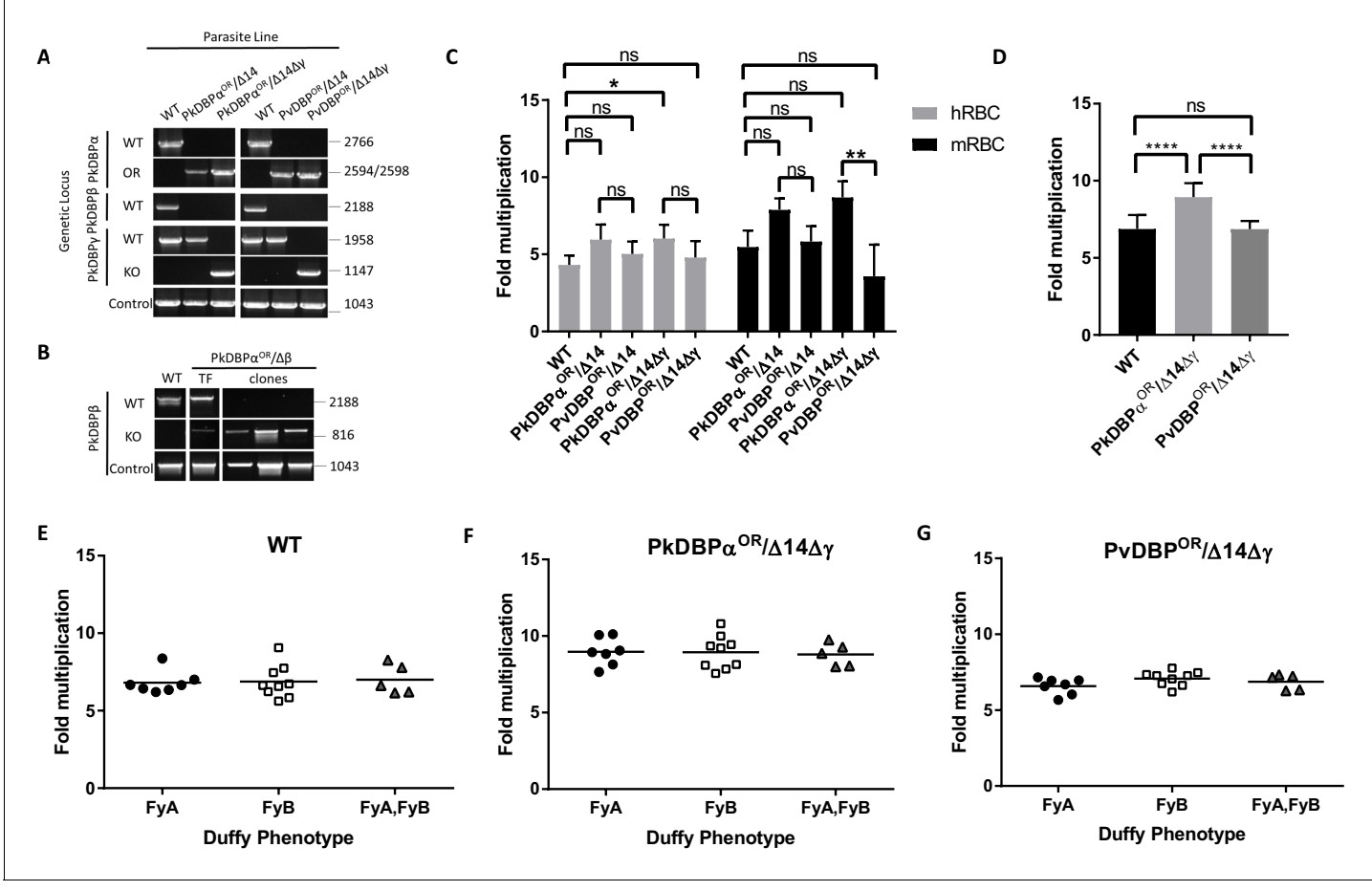

**Figure 4.** PvDBP expressing *P. knowlesi* line demonstrates preference for growth in human RBCs but no preference for different Duffy haplotypes. (**A**) The *P. knowlesi* Duffy binding protein α (DBPα) gene was targeted for replacement with either a recodonised PkDBPα or *P. vivax* DBP repair template. Sequencing revealed a loss of ~44 kb in chromosome 14, including loss of PkDBPβ (PkDBPα^OR/Δ14 and PvDBP^OR/Δ14). These lines were then subsequently modified to knockout PkDBPγ (PkDBPα^OR/Δ14Δγ and PvDBP^OR/Δ14Δγ). Parasite lines were analysed using PCR reactions detecting the wild type (WT) locus PkDBPα (ol186 +ol188), orthologue replacement (OR) locus of PkDBPα^OR (ol186 +ol189) or PvDBP^OR (ol186 +ol187), WT PkDBPβ locus (ol480 +481), WT locus of PkDBPγ (ol483 +ol484), KO locus of PkDBPγ (ol483 +ol258) and a control PCR targeting an unrelated locus (ol75 +ol76). (**B**) The PkDBPα^OR line was modified to knockout PkDBPβ (PkDBPα^OR/Δβ). Parasite lines including the transfection line (TF) and three clones were analysed using PCR reactions detecting WT locus of PkDBPβ (ol480 +ol481), KO locus of PkDBPβ (ol284 +ol481) and a control PCR targeting an unrelated locus (ol75 +ol76). (**C**) Bar chart showing mean fold replication of parasites lines in FACS-based multiplication assays over one growth cycle (24 hr). Assays were carried out in eight biological independent experiments for human blood (hRBC) and three biological independent experiments for *Macaca fascicularis* blood (mRBC). Data points represent mean growth rates and error bars denote ±1 SD. Replication rates of the parasite lines were compared by using one-way ANOVA with Tukey's multiple comparisons test of means. There are significant differences in fold multiplication rates of WT against PkDBPα^OR/Δ14Δγ in hRBCs (p<0.05) and significant differences in fold multiplication rates of PkDBPα^OR/Δ14Δγ against PvDBP^OR/Δ14Δγ in mRBCs (p<0.01). (**D**) Graph showing fold multiplication of WT, PkDBPα^OR/Δ14Δγ and PvDBP^OR/Δ14Δγ *P. knowlesi* parasites in RBC over one intraerythrocytic growth cycle (24 hr). Assays were carried out in technical duplicates in Duffy positive RBC from 21 volunteers with three independent schizont purifications. Data points represent the mean multiplication rate and error bars denote ±1 SD, and were compared by using one-way ANOVA with Tukey's multiple comparisons test of means. There are significant differences in fold multiplication rates of WT against PkDBPα^OR/Δ14Δγ (p<0.001) and PkDBPα^OR/Δ14Δγ against PvDBP^OR/Δ14Δγ (p<0.001). (**E**) Graph showing fold multiplication of WT (**F**) PkDBPα^OR/Δ14Δγ and (**G**) PvDBP^OR/Δ14Δγ *P. knowlesi* parasites in RBC from 21 volunteer blood donors over one intraerythrocytic growth cycle (24 hr). Mean average of fold multiplication rates are plotted against Duffy phenotype [Fy^a, Fy^b, and Fy^(a+b+)]. Black bars indicate mean multiplication rate in each blood type. Data points represent the mean and error bars denote ±1 SD of three biological independent experiments (n = 3); *Figure 4—source data 1*. .ns p>0.05, *<0.05, **<0.01, ***<0.001.

DOI: https://doi.org/10.7554/eLife.45829.013

The following source data and figure supplements are available for figure 4:

**Source data 1.** Source data for graphs.
DOI: https://doi.org/10.7554/eLife.45829.018
**Source data 2.** Primer pairs and vector design for DBP constructs.

*Figure 4 continued on next page*

*Figure 4 continued*

DOI: https://doi.org/10.7554/eLife.45829.019

**Figure supplement 1.** Transgenic *P. knowlesi* DBP orthologue replacement, knockout design and genotypic analysis.

DOI: https://doi.org/10.7554/eLife.45829.014

**Figure supplement 1—source data 1.** Source data for graphs.

DOI: https://doi.org/10.7554/eLife.45829.015

**Figure supplement 2.** Off-target guide sequences for PkDBPα sgRNA.

DOI: https://doi.org/10.7554/eLife.45829.016

**Figure supplement 3.** Sequencing of PvDBP$^{OR}$/Δ14Δγ parasite line.

DOI: https://doi.org/10.7554/eLife.45829.017

mapping against the A1-H.1 reference genome revealed that the PkDBPα$^{OR}$ and PvDBP$^{OR}$ line have a ~ 44 kb truncation at one end of chromosome 14 (*Figure 4—figure supplement 3*), which also harbours DBPβ, therefore we renamed the lines PkDBPα$^{OR}$/Δ14 and PvDBP$^{OR}$/Δ14. The loss of the ~44 kb of chromosome 14 is also present in parasites that have been transfected simultaneously with pCas/sg_*p230p*, suggesting that the 44 kb deletion occurred in the A1-H.1 parental parasite line (wt/Δ14) prior to transfection and was not an artefact caused by targeting DBPα. Similar spontaneous deletions have been reported previously, including ~66 kb loss at the other end of chromosome 14 in the *P. knowlesi* A1-C line maintained in cynomolgus macaque blood that included the invasion ligand NBPXa (*Moon et al., 2016*), and a deletion of DBPγ in the PkYH1 line at the end of chromosome 13 (*Dankwa et al., 2016*). Furthermore, the PAM site of the DBPα targeting guide sequence is absent in DBPβ (*Figure 4—figure supplement 1D*) which makes it unlikely that the disruption of DBPβ was induced by Cas9 during DBPα targeting. To confirm this, another PkDBPα$^{OR}$ clonal line was generated in an independent transfection using the A1-H.1 parental parasite line with intact DBPβ locus.

Having established accurate targeting of the PkDBPα locus, we investigated the role of the paralogues in human and macaque red cell invasion and whether they could interfere with inhibitory effects of test antibodies. Reasoning that as the Δ14 truncation may have provided a selective advantage to the parasites and thus may well reoccur, we took advantage of the spontaneous DBPβ loss (PkDBPα$^{OR}$/Δ14 or PvDBP$^{OR}$/Δ14) and then used pCas9/sg plasmid recycling to additionally delete the DBPγ locus (*Figure 4—figure supplement 1B*), generating PkDBPα$^{OR}$/Δ14Δγ and PvDBP$^{OR}$/Δ14Δγ. An overview of all generated DBP lines is depicted in *Figure 4—figure supplement 1E*).

The final PvDBP$^{OR}$/Δ14Δγ clonal line was subjected to whole genome sequencing to verify changes at all three loci, and this confirmed precise targeting of the PvDBP allele swap into the PkDBPα locus, and complete deletion of the DBPγ open reading frame in the PkDBP γ locus (*Figure 4—figure supplement 3*). Using the PkDBPα$^{OR}$ clonal line with the start of chromosome 14 intact, we were able to generate a Cas9 mediated DBPβ knockout (PkDBPα$^{OR}$/Δβ) (*Figure 4B* and *Figure 4—figure supplement 1C*). FACS-based multiplication assays showed no growth effects of PkDBPβ knockout in human blood (*Figure 4—figure supplement 1F*).

Analysis of the wild type line, and the four transgenic lines (PkDBPα$^{OR}$/Δ14, PvDBP$^{OR}$/Δ14, PkDBPα$^{OR}$/Δ14Δγ and PvDBP$^{OR}$/Δ14Δγ) revealed no difference in growth rate in human RBCs for all lines except the PkDBPα$^{OR}$/Δ14Δγ which demonstrated significantly increased growth rate compared to wild type (*Figure 4C* and *Figure 4—source data 1*). This confirmed that the *P. vivax* protein was able to complement the role of its *P. knowlesi* orthologue. It also demonstrated, that not only are PkDBPβ and γ proteins dispensable for multiplication in humans RBCs (*Dankwa et al., 2016*), but as their loss leads to increased multiplication rate, they may actually impede invasion in human RBCs. A further experiment comparing multiplication rates in 21 Duffy positive blood donations revealed significant higher growth rates of the PkDBPα$^{OR}$/Δ14Δγ line (8.9 fold) compared to the 6.9 fold multiplication of the wild type line and PvDBP$^{OR}$/Δ14Δγ line (p<0.001) (*Figure 4D* and *Figure 4—source data 1*), confirming that knockout of the two paralogues increases multiplication rate in human RBCs but also revealing a minor growth reduction in the PvDBP$^{OR}$/Δ14Δγ line compared to its control line.

To investigate how these modifications affect host preference we compared the growth rates in human RBCs with growth rates in RBCs from the natural host of *P. knowlesi*, the long tailed macaque (*Macaca fascicularis*). Even human culture-adapted *P. knowlesi* retains a strong preference for macaque cells (*Moon et al., 2013*; *Lim et al., 2013*; *Moon et al., 2016*) and it has been

hypothesized that the additional invasion pathways provided by DBPβ and DBPγ are in part responsible for the increased invasion efficiency in macaque RBCs (*Dankwa et al., 2016*). Interestingly, loss of DBPβ and DBPγ in the PkDBPα$^{OR}$/Δ14Δγ line did not reduce the parasite multiplication rate in macaque RBC but rather slightly increased it (*Figure 4C*), with the lines retaining a macaque preference ratio (macaque fold growth/human fold growth) of 1.43, similar to both the wild type (1.26) and PkDBPα$^{OR}$/Δ14 (1.33). This demonstrates that both PkDBPβ and PkDBPγ are dispensable for invasion of macaque RBCs, and PkDBPα is sufficient to retain full growth rate in macaque cells. Unlike *P. knowlesi*, *P. vivax* is unable to infect macaques, and sequence differences between the DARC receptor in the two hosts have been suggested to underlie this restriction (*Tachibana et al., 2015*). Whilst in macaque RBCs the multiplication rates and host preference ratio (1.16) were unaffected for the PvDBP$^{OR}$/Δ14 line, the additional deletion of DBPγ in the PvDBP$^{OR}$/Δ14Δγ line resulted in a 40% reduction in macaque multiplication rates (*Figure 4B*) which caused a shift to human RBC preference with a ratio of 0.75. This suggests that in the absence of redundant DBP pathways, PvDBP is less effective at facilitating invasion of macaque cells than of human cells, but nevertheless can support invasion of both host cell types.

The Duffy gene exists as three distinct alleles, Fy$^a$, Fy$^b$ and Fy$^{null}$. Whilst the Fy null phenotype has long been established to protect individuals from *P. vivax*/*P. knowlesi* infection (*Miller et al., 1975*; *Miller et al., 1976*), previous studies have also shown stronger binding of recombinant PvDBPRII to Fy$^{(a−b+)}$ and Fy$^{(a+b+)}$ human RBCs compared to Fy$^{(a+b−)}$ as well as higher risks of clinical episodes of *P. vivax* malaria in Fy$^{(a−b+)}$. This suggests that *P. vivax* may invade to Fy$^{(a+b−)}$ RBCs less efficiently (*King et al., 2011*; *Fong et al., 2018*; *Kano et al., 2018*). Whilst we have previously been unable to detect any Fy subtype preference in *P. knowlesi* A1-H.1 (22), the PvDBP$^{OR}$ lines provided an opportunity to see if multiplication rate was sensitive to the Duffy subtype when the parasites used PvDBP for invasion. Schizonts of the wild type, PkDBPα$^{OR}$/Δ14Δγ and PvDBP$^{OR}$/Δ14Δγ lines were added in duplicate to washed RBC from 21 volunteers of the three Duffy-positive phenotypes (7x Fy$^a$, 9x Fy$^b$, and 5x Fy$^{(a+b+)}$) at 0.5% parasitaemia. Parasites were maintained in 96-well microtiter plates for 24 hr and the parasitaemia monitored by FACS. For all three lines the average growth rates in each of the three Fy subtypes were similar and there was no significant difference between growth in Fy$^a$ and Fy$^b$ RBCs (*Figure 4E–G*; *Figure 4—source data 1*).

Growth inhibition activity (GIA) assays revealed that all lines remained equally susceptible to invasion inhibition by both an anti-DARC camelid nanobody CA111 and a polyclonal αPkMSP1$_{19}$ antibody (*Figure 5A*). In contrast, purified IgGs from polyclonal rabbit sera raised against PvDBP_RII, demonstrated low-level GIA activity for wild type and PkDBPa$^{OR}$/ΔβΔγ lines (~30% inhibition at 10 mg/ml) but a significantly stronger GIA activity against the PvDBP$^{OR}$/Δ14 and PvDBP$^{OR}$/ΔβΔγ lines, reaching a maximum inhibition of ~75% at 10 mg/ml and around 50% at 4 mg/ml (*Figure 5B and C*). At 2.5 mg/ml IgG the GIA activity against the PvDBP$^{OR}$/Δ14 and PvDBP$^{OR}$/ΔβΔγ lines were 29.6 ± 3.3 and 24.6 ± 2.6% and significantly higher compared to wild type and PkDBPa$^{OR}$/ΔβΔγ lines (p<0.001) (*Figure 5D and E*). The PvDBP$^{OR}$ parasite lines could thus be readily inhibited by antibodies against the *P. vivax* protein and the PkDBPα orthologues PkDBPγ and β appeared to play no interactive role.

We thus have created a transgenic *P. knowlesi* model, modified at two separate loci which recapitulates the *P. vivax* DBP invasion pathway. This parasite line is a vital new tool in PvDBP vaccine development.

## Discussion

In this work, we adapt CRISPR-Cas9 genome editing to the zoonotic malaria parasite *P. knowlesi*. Whilst various approaches for CRISPR-Cas9 have been used for other malaria parasites, here we combine a plasmid containing a single recyclable positive selection marker with a fusion PCR-based approach for generation of repair templates. This allows for seamless insertion or deletion at any location within a gene and unlimited iterative modifications of the genome. Genome-wide reverse genetics screens have been applied with great success to the rodent malaria parasite, *P. berghei*, but they have remained challenging for *P. falciparum*, and impossible for *P. vivax*. The tools presented here will enable scalable construct assembly and genome-wide systematic knockout or tagging screens in an alternative human infective species, thus providing a complementary tool to address both shared and species-specific biology. The analysis of lines with multiple tagged or

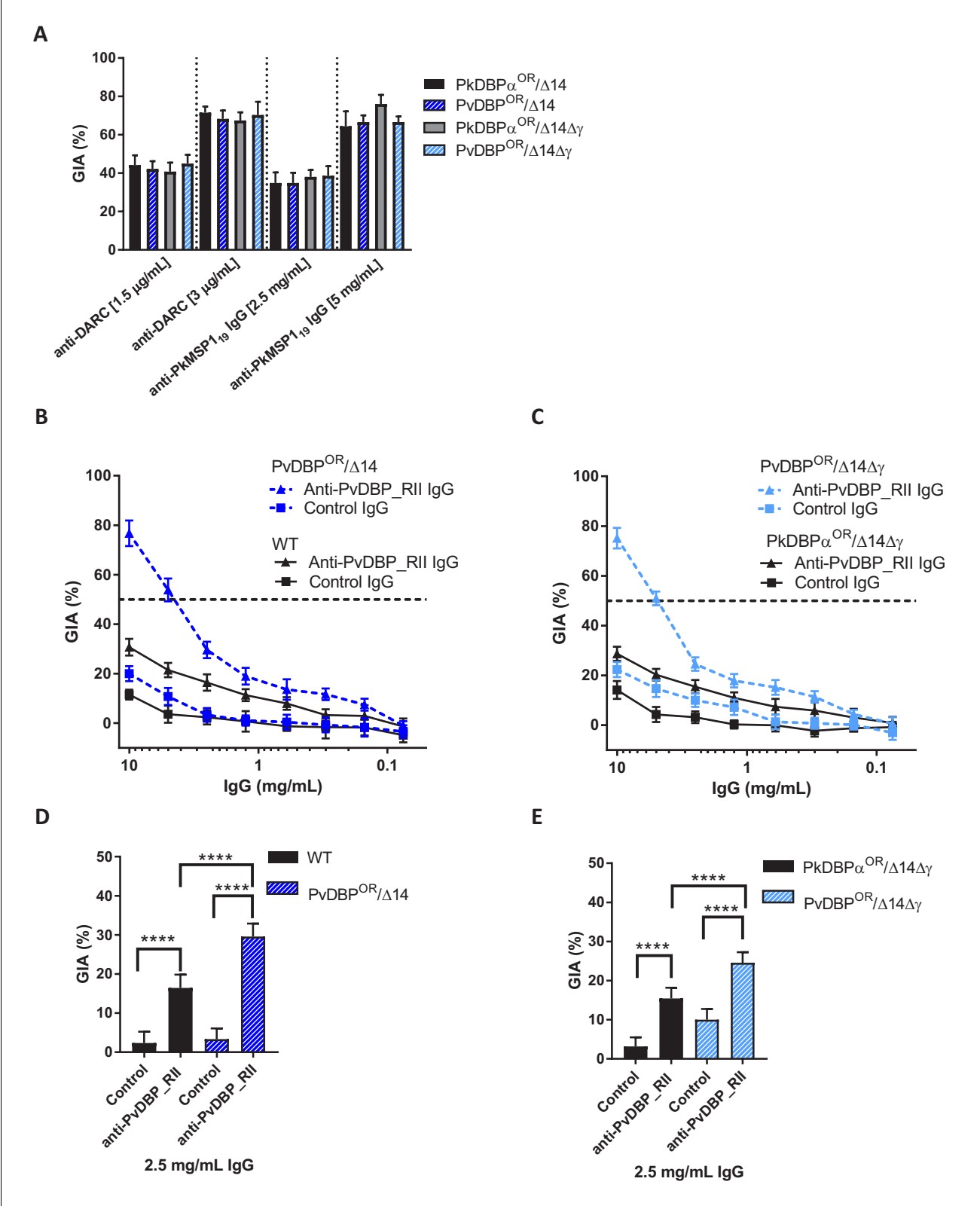

**Figure 5.** Transgenic *P. knowlesi* orthologue replacement lines provide surrogates for *P. vivax* vaccine development. (**A**) Graph showing growth inhibition activity (GIA, %) of anti-DARC nanobody at 1.5 and 3 µg/ml and anti-MSP1$_{19}$ purified total rabbit IgG at 2.5 and 5 mg/ml on the parasite lines. Data points represent the mean and error bars denote ±1 SD of triplicate test wells (n = 3). GIAs of each antibody were compared across the parasite lines by using unpaired one-way ANOVA with Tukey's multiple comparisons test of means. No significant changes were observed. (**B**) Graph shows the

*Figure 5 continued on next page*

*Figure 5 continued*

% GIA of a dilution series of IgG purified from sera of PvDBP_RII (SalI)-immunized rabbits as well as control IgG from the pre-immunisation sera of the same rabbits against wild type (WT) and PvDBP$^{OR}$/Δ14 transgenic *P. knowlesi* lines and (C) against PkDBPα$^{OR}$/Δ14Δγ and PvDBP$^{OR}$/Δ14Δγ lines. Data points represent the mean and error bars denote ±1 SD of five or six replicates. (D) Bar chart showing % GIA of 2.5 mg/ml IgG purified from sera of PvDBPRII (SalI)-immunized rabbits as well as control IgG from the pre-immunisation sera of the same rabbits against wild type (WT) and PvDBP$^{OR}$/Δ14 transgenic *P. knowlesi* lines and (E) against PkDBPα$^{OR}$/Δ14Δγ and PvDBP$^{OR}$/Δ14Δγ lines. Bars represent the mean and error bars denote ±1 SD of five or six replicates and were compared by using one-way ANOVA with Tukey's multiple comparisons test of means. ns p>0.05, *<0.05, **<0.01, ***<0.001.
DOI: https://doi.org/10.7554/eLife.45829.020

The following source data is available for figure 5:

**Source data 1.** Source data for graphs.
DOI: https://doi.org/10.7554/eLife.45829.021
**Source data 2.** Full primer list for entire study.
DOI: https://doi.org/10.7554/eLife.45829.022

deleted genes is particularly valuable for multigene families with highly redundant functions, as exemplified by our modification of all three *P. knowlesi* DBP genes.

Here we investigate key parameters associated with successful genome editing and show that the process is also highly robust; targeting of the *p230p* locus demonstrated successful editing for 25/25 transfections and only 1/10 sgRNAs targeting different loci failed to generate an edited line. The failure of an sgRNA guide (AGAAAATAGTGAAAACCCAT) designed to target the DBPβ locus, a non-essential gene, suggests that multiple guides may need to be tested for some loci. All sgRNA guides used are shown in *Figure 3—source data 1*. We did not detect any off-target effects, consistent with other reports of CRISPR-Cas9 use in malaria parasites (*Ghorbal et al., 2014*; *Lu et al., 2016*; *Wagner et al., 2014*; *Zhang et al., 2017*). Negative selection of the pCas9/sg plasmid then enables generation of markerless lines allowing unlimited iterative modifications of the genome, with each round requiring only ~30 days (including dilution cloning). Whilst the generation of markerless Cas9 modifications in the genome of *P. falciparum* has been possible for some time, initial systems contained multiple different selectable markers and relied on passive loss of episomes to recycle at least one marker (*Ghorbal et al., 2014*; *Mogollon et al., 2016*; *Lu et al., 2016*; *Wagner et al., 2014*). Whilst this is possible in some cases, it can create difficulties particularly if long selection protocols are necessary which may result in stabilisation/integration of episomes. Our format is similar to a more recent system developed in *P. falciparum* (*Knuepfer et al., 2017*), which by using only one bifunctional positive/negative selection cassette enables complete recycling of selectable markers. This has critically enabled generation of important base lines such as marker-free dimerisable Cre recombinase lines used for conditional knockouts in *P. falciparum* (*Knuepfer et al., 2017*). More extensive modifications using Cas9, such as the three sequential genetic modifications undertaken for *P. yoelli* (*Zhang et al., 2017*), remains relatively uncommon suggesting that the timescales involved in generation of these lines is still a challenge.

One clear disadvantage of this system is that we do still see a background remnant of wild type parasites for many of the modifications that are made. This may mean that modifications resulting in slow growing phenotypes could create a challenge for generating clonal lines. In our subsequent experience this is largely mitigated by the relatively fast grow out times and in extreme instances may be combated by immediately cloning after transfection. The selection-linked integration (SLI) method used in *P. falciparum* and *P. knowlesi* rapidly selects for genomic integration by using an additional selectable marker that is only expressed when correctly integrated, resulting in transgenic lines that do not require cloning (*Birnbaum et al., 2017*; *Lyth et al., 2018*). The disadvantage of this approach is that multiple selectable markers are required and at least one of them cannot be recycled in the final transgenic line, limiting scope for subsequent modifications. In our experience, the ability to extensively and iteratively modify parasites has been perhaps the most transformative aspect of this method and was integral to our work to modify all members of the PkDBP family. Importantly, as Cas9 modified lines are markerless, the two systems are entirely compatible, such that a baseline containing a range of Cas9 induced modifications could then be modified again using a SLI-based approach to enable access to more challenging slow growing mutants.

Parasites with integration of eGFP into the p230p locus reached 1% parasitaemia in 8 to 14 days after transfection. However for the most efficient transfections we were able to observe parasites

reaching 1% at only 5 to 6 days suggesting significant differences in targeting efficiency at different loci and with different gRNAs. Whilst *P. knowlesi* grows slightly faster (with 3–4 fold per cycle equating to around 9-fold in 48 hr) than *P. falciparum*, this still indicates orders of magnitude greater modification rates than achieved for *P. falciparum* - which can take 2–6 weeks for transgenic parasites to be detected in culture depending on the transfection system used (*Ghorbal et al., 2014*; *Mogollon et al., 2016*; *Lu et al., 2016*; *Knuepfer et al., 2017*; *Wagner et al., 2014*). Despite the relatively fast recovery time it is also clear that as more than 8% of parasites are able to take up donor DNA during transfection it still takes more than a week for integrated parasites to grow out, indicating that efficiency of homology mediated repair is still the major bottleneck and the majority of Cas9 induced DSBs are not successfully repaired.

We systematically tested key parameters associated with successful genome editing and found increasing HR length enhanced integration efficiency proportionately, a trend seen in both *P. falciparum* and *P. berghei* (*Collins et al., 2013*; *Wagner et al., 2014*; *MacPherson and Scherf, 2015*). Whilst integration was detected with HRs as short as 50 bp, efficient editing was achieved with HRs between 200–800 bp. We were also able to examine how distance from the DSB affected editing efficiency. Whilst in other systems editing efficiency decreases rapidly as the DSB distance increases, we saw only a steady decline with distance, an effect which could be ameliorated by simply increasing HR length. The use of PCR products as repair templates is particularly well adapted for tagging and knockout approaches which tend to be relatively short. For relatively large or complex constructs, such as those for gene replacement, or those targeting multigenic families conventional cloning may be preferable, as shown here for the PkDBP family.

By applying these techniques to the *P. knowlesi* and *P. vivax* DBP family we have been able to examine the role of these genes in host and reticulocyte preference of the two species. Even after long-term adaptation to culture with human RBCs, *P. knowlesi* parasites can retain a strong preference for invasion of macaque RBCs (*Dankwa et al., 2016*; *Moon et al., 2016*). Both DBPγ and DBPβ have been shown to bind to proteins with a distinct sialic acid residue found in non-human primates, but absent in humans (*Dankwa et al., 2016*). Deletion of these genes led to a significant increase in growth rates in human RBCs, and interestingly also a similar, but non-significant, increase in growth rates in macaque RBCs. This suggests that PkDBPα alone is sufficient to retain the multiplication capacity and that the PkDBP paralogues are not responsible for the macaque cell preference retained in the A1-H.1 human adapted line. Whilst likely non-functional for invasion of human RBCs, both PkDBPγ and DBPβ would still compete for surface space and potentially some of the same interacting proteins as DBPα, which may account for the increased growth rate when they are deleted. The increased growth rate in macaque cells, would suggest that even here, where DBPγ and DBPβ are functional, PkDBPα is the preferred and more efficient pathway for invasion in macaque cells. In vivo, this advantage would be countered by the significant benefits provided by redundancy both to combat host blood cell polymorphisms and antibody responses. Importantly, this data suggests that loss of DBPγ and DBPβ could provide an adaptation route to increased growth/virulence within human infections. However, to do so the parasites would need to sacrifice redundancy within its primary macaque hosts – a situation only likely to occur with prolonged human to human transmission. Further work to analyse the effect of these mutations through long-term parasite competition assays will enable us to determine the precise extent of this advantage and determine how quickly such mutations could move to fixation within a parasite population.

Despite being closely related to *P. knowlesi* and other macaque infecting species such as *P. cynomolgi*, *P. vivax* cannot infect macaques and the PvDBP protein has been suggested to play a role in enforcing this tropism, as key interacting residues are missing within the macaque DARC protein (*Tachibana et al., 2015*). *P. knowlesi* parasites expressing PvDBP in the absence of DBP paralogues demonstrate a significant reduction in growth in macaque cells, resulting in an overall shift towards preference for human cells consistent with PvDBP binding macaque DARC less efficiently. Nevertheless as the multiplication rate remained quite close to that seen for human RBCs it seems unlikely that the PvDBP protein alone represents a significant barrier to *P. vivax* infection of macaques.

Previous work has shown that PvDBP_RII binds more strongly to RBCs expressing the Fy[b] allele of the Duffy antigen than Fy[a], with studies using recombinant protein demonstrating that the binding efficiency is 50% lower for Fy[a] (*Kano et al., 2018*). This effect was shown to correlate with a 30–80% reduced risk of clinical vivax malaria amongst patients with the FyA phenotype (*King et al., 2011*). A similar effect of reduced DBP-FyA binding has also been demonstrated using recombinant PkDBPα

(*Fong et al., 2018*) but experiments to examine this with human RBC culture adapted *P. knowlesi* parasites failed to demonstrate any difference in multiplication rates in either $Fy^{b+}$ or $Fy^{a+}$ RBC (*Moon et al., 2013*). Here, we also did not observe a significant association between growth of the PvDBP$^{OR}$/$\Delta14\Delta\gamma$ line and Duffy haplotype. Whilst the size of our dataset cannot rule out relatively subtle effects on parasite growth rate, which over the course of an infection could become clinically relevant, it does rule out major growth differences and demonstrates that the invasion process is very tolerant of even quite large changes in receptor-ligand affinity. Notably, the study first identifying a role for Duffy polymorphisms in *P. vivax* clinical susceptibility also demonstrated that the $Fy^{a+}$ RBCs demonstrate increased sensitivity to invasion-blocking antibodies, so protective effects may stem from this rather than a direct effect on invasion efficiency (*King et al., 2011*; *Moon et al., 2013*). Subsequent experiments using these lines could readily test this hypothesis.

Another key difference between the two species is that unlike *P. knowlesi*, *P. vivax* has a strict restriction to invasion of reticulocytes. A second family of RBC binding proteins, known as the reticulocyte binding-like proteins (RBPs) have previously been implicated in this tropism. More recently, the PvDBP protein itself has been implicated with work using recombinant PvDBP_RII suggesting that whilst DARC is present on both reticulocytes and mature normocytes, changes during red cell maturation mean that DARC is only accessible to PvDBP binding in young reticulocytes (*Ovchynnikova et al., 2017*; *Haynes et al., 1988*). Here we show that transgenic *P. knowlesi* parasites using PvDBP for invasion have no such restriction, invading human RBCs (which typically contain less than 0.5% reticulocytes) with the same efficiency as those expressing PkDBP$\alpha$ – thus providing compelling evidence that PvDBP plays no role in the reticulocyte tropism. Further, recent work determining that PvRBP2b, which lacks an orthologue in *P. knowlesi*, binds to the reticulocyte specific marker CD71 further asserts the RBPs as the key to reticulocyte tropism. Importantly, the ability to compare and contrast activity of Pk/Pv DBP family members in parasitological assays will provide a vital new tool to test hypotheses and models arising from studies that have until now relied on assays using recombinant protein fragments.

Using a combination of binding assays, mutagenesis and structural studies with recombinant PvDBP_RII, previous work has identified residues involved in DBP-DARC interactions (*Choe et al., 2005*; *Hans et al., 2005*; *VanBuskirk et al., 2004*), as well as a DBP dimerization domain thought to drive the stepwise engagement with DARC (*Batchelor et al., 2014*; *Batchelor et al., 2011*). Efforts to develop a *P. vivax* vaccine to elicit antibodies against the lead candidate PvDBP have predominantly relied on using ELISA-based assays, which assess the ability of antibodies to block recombinant PvDBP_RII binding to DARC (*Shakri et al., 2012*). This has successfully identified a range of binding inhibitory antibodies mapped to residues involved in DARC binding, dimerization as well as subdomain 3, which is distant from the DARC binding site, demonstrating that multiple mechanisms of inhibition may be important (*Chen et al., 2016*; *Chootong et al., 2010*). However, validation of these findings in a parasitological assay is critical. Some epitopes identified in recombinant protein assays may be inaccessible in the context of invasion and it is also possible that not all inhibitory antibodies block receptor binding. DARC-DBP binding is only one step in the multi-step invasion process, with subsequent conformational changes and potential downstream signalling roles for the protein (*Batchelor et al., 2011*). The full-length DBP antigen is 140 kDa which contains a C-terminal transmembrane domain and as such structural and biochemical analysis of the protein has almost exclusively focused on the PvDBP_RII fragment alone. Whilst efforts to standardise ex vivo *P. vivax* assays have been successful (*Russell et al., 2011*), they remain hugely challenging, low throughput and rely on genetically diverse *P. vivax* clinical isolates, that are maintained in culture for a only a single cycle of RBC invasion. The *P. knowlesi* PvDBP$^{OR}$ line thus provides an opportunity to interrogate the function of the full-length protein in an accessible and scalable manner.

Whilst the relationship between in vitro GIA and in vivo protection against malaria challenge has not always been clear, previous studies have shown that there is a strong correlative link between activity in the in vitro assay of GIA and in vivo protection from challenge of two different species of human malaria (*P. falciparum* and *P. knowlesi*) in two different non-human primate models (Aotus monkeys and rhesus macaques) targeting three different merozoite proteins (PfAMA1, PfRH5, PfMSP1) (*Douglas et al., 2015*; *Mahdi Abdel Hamid et al., 2011*; *Singh et al., 2006*). Importantly, Douglas et al. in 2015 determined a threshold value for in vitro GIA assay which must be reached before protection from malaria challenge is afforded (*Douglas et al., 2015*). Thus, the ability to

readily assess *P. vivax* candidates in an in vitro GIA offers an important mechanism to determine whether candidates are appropriate for moving on to challenge studies.

A vaccine against *P. vivax* must ultimately elicit antibodies with strain-transcending inhibitory activity, but the ability to test on a defined genetic background can provide a significant advantage when it comes to benchmarking and prioritising target epitopes and characterising sera raised against them. Here we use the PvDBP sequence from the Sall reference strain, but multiple lines expressing distinct PvDBP variants could be generated in future, as has already been used for recombinant ELISA based assays (*Ntumngia et al., 2012*), to systematically examine inhibition in heterologous strains. Isolates refractory to a given test antibody in ex vivo assays can then be sequenced and direct the generation of new transgenic *P. knowlesi* PvDBP$^{OR}$ variant lines to support rational vaccine development. These assays in turn can provide vital triaging for non-human primate models, and controlled human challenge infections (*Payne et al., 2017a*; *Russell et al., 2011*; *Arévalo-Herrera et al., 2005*; *Payne et al., 2017b*) – both of which carry the imperative to ensure that only highly optimised antigens are tested. The transgenic *P. knowlesi* OR lines developed here represent the ideal platform for scalable testing of polyclonal and monoclonal sera from vaccine trials and natural *P. vivax* infections. This will enable detailed investigation of epitopes providing invasion inhibitory activity and a means for systematic development of a strain-transcending vaccine. Our work also revealed low-level cross-reactivity of PvDBP_RII antibodies against *P. knowlesi* and suggests cross-immunity between the two species could exist in the field, which may have a significant impact on disease outcome. Understanding the precise epitopes involved could facilitate development of a dual species vaccine, and epitopes conserved across species are also more likely to be conserved across polymorphic strains of *P. vivax*. The same approach could readily be applied to other potential vaccine candidates, novel drug targets or to investigate mechanisms of drug resistance, which are also thought to differ between *P. falciparum* and *P. vivax* (*Price et al., 2014*).

In conclusion, we demonstrate that adaptation of CRISPR-Cas9 genome editing to *P. knowlesi* provides a powerful system for scalable genome editing of malaria parasites and can provide critical new tools for studying both shared and species-specific biology.

# Materials and methods

## Key resources table

| Reagent type (species) or resource | Designation | Source or reference | Identifiers | Additional information |
|---|---|---|---|---|
| Cell line (*Plasmodium knowlesi*) | A1-H.1 wild type (WT) | (*Moon et al., 2013*), Mike Blackman, Francis Crick Institute London | | |
| Cell line (*Plasmodium knowlesi*) | p230p eGFP cassette | this paper | | Can be obtained from Rob Moon, LSHTM |
| Cell line (*Plasmodium knowlesi*) | AMA1-HA | this paper | | Can be obtained from Rob Moon, LSHTM |
| Cell line (*Plasmodium knowlesi*) | RON2-HA | this paper | | Can be obtained from Rob Moon, LSHTM |
| Cell line (*Plasmodium knowlesi*) | Myosin A-eGFP | this paper | | Can be obtained from Rob Moon, LSHTM |
| Cell line (*Plasmodium knowlesi*) | CRT-eGFP | this paper | | Can be obtained from Rob Moon, LSHTM |
| Cell line (*Plasmodium knowlesi*) | mCherry-K13 | this paper | | Can be obtained from Rob Moon, LSHTM |

*Continued on next page*

*Continued*

| Reagent type (species) or resource | Designation | Source or reference | Identifiers | Additional information |
|---|---|---|---|---|
| Cell line (*Plasmodium knowlesi*) | PkDBPα$^{OR}$/Δ14 | this paper | | Can be obtained from Rob Moon, LSHTM |
| Cell line (*Plasmodium knowlesi*) | PvDBP$^{OR}$/Δ14 | this paper | | Can be obtained from Rob Moon, LSHTM |
| Cell line (*Plasmodium knowlesi*) | PkDBPα$^{OR}$/Δ14Δγ | this paper | | Can be obtained from Rob Moon, LSHTM |
| Cell line (*Plasmodium knowlesi*) | PvDBP$^{OR}$/Δ14Δγ | this paper | | Can be obtained from Rob Moon, LSHTM |
| Cell line (*Plasmodium knowlesi*) | PkDBPα$^{OR}$ | this paper | | Can be obtained from Rob Moon, LSHTM |
| Cell line (*Plasmodium knowlesi*) | PkDBPα$^{OR}$/Δβ | this paper | | Can be obtained from Rob Moon, LSHTM |
| Transfected construct (plasmid) | pCas9/sg_p230p | this paper | | Can be obtained from Rob Moon, LSHTM |
| Transfected construct (plasmid) | pCas9/sg_PkDBPα | this paper | | Can be obtained from Rob Moon, LSHTM |
| Transfected construct (plasmid) | pCas9/sg_DBPβ | this paper | | Can be obtained from Rob Moon, LSHTM |
| Transfected construct (plasmid) | pCas9/sg_DBPγ | this paper | | Can be obtained from Rob Moon, LSHTM |
| Transfected construct (plasmid) | pCas9/sg_AMA1 | this paper | | Can be obtained from Rob Moon, LSHTM |
| Transfected construct (plasmid) | pCas9/sg_RON2 | this paper | | Can be obtained from Rob Moon, LSHTM |
| Transfected construct (plasmid) | pCas9/sg_Myosin A | this paper | | Can be obtained from Rob Moon, LSHTM |
| Transfected construct (plasmid) | pCas9/sg_K13 | this paper | | Can be obtained from Rob Moon, LSHTM |
| Transfected construct (plasmid) | pCas9/sg_CRT | this paper | | Can be obtained from Rob Moon, LSHTM |
| Transfected construct (plasmid) | pDonor_p230p | this paper | | Can be obtained from Rob Moon, LSHTM |
| Transfected construct (plasmid) | pDonor_PkDBPα$^{OR}$ | this paper | | Can be obtained from Rob Moon, LSHTM |
| Transfected construct (plasmid) | pDonor_PvDBP$^{OR}$ | this paper | | Can be obtained from Rob Moon, LSHTM |

*Continued on next page*

*Continued*

| Reagent type (species) or resource | Designation | Source or reference | Identifiers | Additional information |
|---|---|---|---|---|
| Transfected construct (plasmid) | pDonor_PkDBPβ | this paper | | Can be obtained from Rob Moon, LSHTM |
| Transfected construct (plasmid) | pDonor_PkDBPγ | this paper | | Can be obtained from Rob Moon, LSHTM |
| Antibody | αPkMSP1$_{19}$ (rabbit polyclonal) | Ellen Knuepfer, Francis Crick Institute London | | 2.5 and 5 mg/ml |
| Antibody | anti-DARC nanobody CA111 (camel) | Olivier Bertrand, INSERM, France (*Smolarek et al., 2010*) | | 1.5 and 3 µg/ml |
| Antibody | PvDBP_RII (rabbit polyclonal) | Simon Draper, Jenner Institute Oxford, (*de Cassan et al., 2015*) | | |
| Antibody | Alexa Fluor(TM) 594 Goat Anti-Rabbit IgG (H + L) highly cross-adsorbed (goat polyclonal) | Invitrogen | Cat. #: A-11037 | dilution 1:5000 |
| Antibody | anti-HA high affinity (3F10), 50 UG (rat monoclonal) | Sigma | Cat. #: 11867423001 | dilution 1:5000 |
| Antibody | anti-mCherry (rabbit polyclonal) | Abcam | Cat. #: ab183628 | dilution 1:5000 |
| Antibody | anti-GFP (mouse monoclonal) | Sigma | Cat. #: 11814460001 | dilution 1:5000 |
| Antibody | mouse HRP-conjugated secondary antibody (goat) | Bio-Rad | Cat. #: 1706516 | dilution 1:5000 |
| Antibody | Duoclone Monoclonal | Lorne | Cat. #: 740010 | |
| Antibody | Anti-Human IgG (clear) | Lorne | Cat. #: 401010 | |
| Antibody | Anti-FyB Monoclonal | Lorne | Cat. #: 317002 | |
| Antibody | Anti-FyA Monoclonal | Lorne | Cat. #: 774002 | |
| Antibody | Anti-B Monoclonal | Lorne | Cat. #: 610010 | |
| Antibody | Anti-A Monoclonal | Lorne | Cat. #: 600010 | |
| Chemical compound, drug | 4-[7-[(dimethylamino) methyl]−2-(4-fluorphenyl) imidazo[1,2-a]pyridin-3 -yl]pyrimidin-2-amine (compound 2) | Michael Blackman, Francis Crick Institute London | | |
| Software, algorithm | Protospacer software | http://www.protospacer.com/ | | |
| Software, algorithm | GraphPad Prism | GraphPad Prism (http://graphpad.com) | RRID:SCR_015807 | Version 7 |
| Software, algorithm | Nikon Elements Advanced Research software package | https://www.microscope.healthcare.nikon.com/products/software/nis-elements | | |
| Software, algorithm | Benchling Software | https://www.benchling.com/ | RRID:SCR_013955 | |

*Continued on next page*

*Continued*

| Reagent type (species) or resource | Designation | Source or reference | Identifiers | Additional information |
|---|---|---|---|---|
| Software, algorithm | FACSDiva 6.1.3 software | http://www.bdbiosciences.com/ca/instruments/clinical/software/flow-cytometry-acquisition/bd-facsdiva-software/bd-facsdiva-software-v-613/p/643629 | RRID:SCR_001456 | |
| Software, algorithm | FlowJo_V10 | https://www.flowjo.com/ | RRID:SCR_008520 | version 10 |
| Software, algorithm | *sambamba* software | https://github.com/biod/sambamba | | |
| Software, algorithm | R | https://www.r-project.org/ | RRID: SCR_001905 | |
| Commercial assay or kit | QIAseq FX DNA Library Kit | Quiagen | Cat. #: 180473 | |
| Commercial assay or kit | Pierce Protein A IgG Purification Kit | ThermoFisher Scientific | Cat. #: 44667 | |
| Commercial assay or kit | Recombinant Protein G Agarose | ThermoFisher Scientific | Cat. #: 15920010 | |

## Macaque and human RBCs

*Macaca fascicularis* blood was collected by venous puncture. Animal work was reviewed and approved by the local National Institute for Biological Standards and Control Animal Welfare and Ethical Review Body (the Institutional Review Board) and by the United Kingdom Home Office as governed by United Kingdom law under the Animals (Scientific Procedures) Act 1986. Animals were handled in strict accordance with the 'Code of Practice Part one for the housing and care of animals (21/03/05)' available at https://www.gov.uk/research-and-testing-using-animals. The work also met the National Centre for the Replacement Refinement and Reduction of Animals in Research (NC3Rs) guidelines on primate accommodation, care, and use (https://www.nc3rs.org.uk/non-human-primate-accommodation-care-and-use), which exceed the legal minimum standards required by the United Kingdom Animals (Scientific Procedures) Act 1986, associated Codes of Practice, and the US Institute for Laboratory Animal Research Guide. For most experiments and routine parasite maintenance, human blood (Duffy (FY) positive) was obtained from the United Kingdom National Blood Transfusion Service under a research agreement. For the Duffy subtype experiment, venous blood (~10 mL) from 21 healthy volunteers blood donors was collected into EDTA Vacutainer (SLS), after donors had provided informed consent. Blood samples were anonymized, and then RBCs washed with RPMI 1640 and stored at 4°C. All. The project, consent and protocol were approved by the LSHTM Observational Research Ethics Committee under project reference 5520–1.

## Parasite maintenance, transfection and dilution cloning

Parasites were maintained in complete media, comprising RPMI 1640 (Invitrogen) with the following additions: 2.3 g/L sodium bicarbonate, 4 g/L dextrose, 5.957 g/L HEPES, 0.05 g/L hypoxanthine, 5 g/L Albumax II, 0.025 g/L gentamycin sulfate, 0.292 g/L L-glutamine, and 10% (vol/vol) horse serum as described previously (*Moon et al., 2016*). Parasites were synchronized by using gradient centrifugation with 55% nycodenz (Progen) in RPMI to enrich schizonts, followed by a two-hour incubation with 4-[7-[(dimethylamino)methyl]−2-(4-fluorphenyl)imidazo[1,2-*a*]pyridin-3-yl]pyrimidin-2-amine (compound 2) which inhibits parasite egress (*Collins et al., 2013*). Incubations in compound 2 longer than 2 hr led to degeneration of schizonts and reduction in invasive capacity.

Tightly synchronized mature schizonts were transfected as described previously using the Amaxa 4D electroporator (Lonza) and the P3 Primary cell 4D Nucleofector X Kit L (Lonza)(*Moon et al., 2013*). 10 µl DNA including at 20 µg repair template pDonor_*p230p* and 20 µg pCas9/sg_*p230p* plasmid was used for transfections to generate eGFP expressing lines. 10 µl DNA including 15 µg repair template and 7 µg pCas9/sg_*p230p* plasmid was used for transfections to integrate the eGFP expression cassette into the *p230p* locus with PCR repair templates. For generating tagged lines 10

µg pCas/sg_GOI plasmid and 20 µg PCR repair templates were used. After 24 hr, and at daily intervals for 5 days, the medium was replaced with fresh medium containing 100 nM pyrimethamine (Sigma). Parasites were cloned out by limiting dilution. Parasites were diluted to 0.3 parasites/100 µl and 100 µl of 2% haematocrit culture was transferred to 96 flat-bottom plates in culture medium containing 200 mM L-glutamax (Sigma). After 7 days the media was changed and 0.2% fresh blood added. On day 11 the plate was screened for plaques, in an assay modified from *P. falciparum* (*Thomas et al., 2016*). Plaque positive cultures were transferred to 24 well plates containing 1 ml media with 2% haematocrit and used for genotyping.

## DNA constructs and PCRs

Preparative DNA for plasmid cloning and PCR fusion constructs was amplified with CloneAmp (Takara) using the following cycle conditions: 32 cycles of 5 s at 98°C, 20 s at 55°C, and 5 s/kb at 72°C. Genomic DNA was prepared using DNeasy blood and tissue kit (Qiagen).

### Cloning of pkcon_mCherry plasmid

The plasmid pkconGFP (*Moon et al., 2013*) was modified to replace the GFP coding sequence with mCherry using XmaI and SacII restriction sites. The mCherry sequence was amplified with primers fwd-ATAT<u>CCCGGG</u>ATGGTGAGCAAGGGCGAGGAG and rev-ATAT<u>CCGCGG</u>TTACTTGTACAGCTCGTCCATGCC.

### Cloning of pCas9/sg

The pUF1 plasmid (*Ghorbal et al., 2014*) was modified by replacing the yDHODH expression cassette with hDHFR-yFCU fusion with PkEF1a 5'UTR and Pbdhfr 3'UTR using EcoRI and SacII. The PfU6 promoter for gRNA expression of the pL6 plasmid (*Ghorbal et al., 2014*) was replaced with the PkU6 5' regulatable region of 1244 bp (amplified with primers fwd-ATATCCATGGGGGCCAGGGAA-GAACGGTTAGAG and rev-atattcgcgagcgatgagttcctaggAATAATATACTGTAAC) using NruI and NcoI and the entire cassette inserted into the pCas9 plasmid with PvuI and ApaI restriction sites. Each target specific 20 bp guide sequence was chosen with the Protospacer software (http://www.protospacer.com/), with off-target score <0.03. On-target scores were retrieved from Benchling Software (*Benchling, 2018*). All guide sequences are listed in *Figure 4—source data 2*. Subsequently each guide was inserted into the BtgZI linearized pCas9/sg plasmid by In-Fusion cloning (Takara) using primers fwd-TTACAGTATATTATT(N20)GTTTTAGAGCTAGAA and rev-TTCTAGCTCTAAAAC(N20)AATAATATACTGTAA. Briefly, 50 bp primer pairs containing the 20 bp guide sequence flanked by 15 bp overhangs homologous to the 5' and 3' ends of pCas9/sg were denatured by incubation at 95°C for 10 min and annealed by slow cooling. 0.5 µM annealed primers and 50 ng BtgZI linearized pCas/sg vector were incubated with In-fusion Premix (Takarta) at 50°C for 15 min. The resulting plasmid was transformed into XL10 gold competent *E. coli* cells (Agilent). Plasmids for transfection were prepared by Midi-preps (QIAGEN) and ethanol precipitated. The DNA pellet was washed twice with 70% ethanol and resuspended in sterile TE buffer.

### Cloning of pDonor_p230p

A plasmid containing a multiple cloning site with SacII, SpeI, NotI, BlpI and NcoI was designed (subsequently called pDonor) and obtained from Geneart (Thermo Fisher Scientific). Homology region 1 (HR1) was amplified from A1-H.1 wild type genomic DNA with primers olFM007 and olFM008 and added with SacII and SpeI restriction sites. HR2 was amplified with primers olFM005 and olFM006 and was added with BlpI and NcoI sites. The eGFP cassette was amplified from the pkconGFP*p230p* plasmid with primers olFM151and olFM152 and inserted into the plasmid with SpeI and BlpI sites. The final vector was linearised with PvuI restriction enzyme and ethanol precipitated as described above.

### Cloning of pDonor_pkdbpα

Plasmid pDonor was modified by restriction cloning to include two 500 bp HRs from PkDBPa 5' and 3'UTRs using primers olFM062 and olFM063 (adding SacII/SpeI sites) and primers olFM064 and olFM065 (adding NotI/NcoI sites) respectively. Recodonised sequences of PkDBPα and PvDBP of the *P. vivax* Salvador I strain flanked with SpeI and NcoI restriction sites were obtained from Geneart

(Thermo Fisher Scientific) and subsequently cloned between both HRs of the modified pDonor plasmid using SpeI and NcoI sites. The resulting plasmid was linearised with PvuI restriction enzyme and ethanol precipitated as described above. Primer pairs are shown in *Figure 4—source data 2* and primer sequences in *Figure 5—source data 2*.

## Cloning of pDonor_ *pkdbpγ*

Plasmid pDonor was modified by restriction cloning to include two HRs from PkDBPγ 5' and 3'UTRs using primers olFM245 and olFM0246 (adding SacII/SpeI sites) and primers olFM0247 and olFM248 (adding NotI/NcoI sites) respectively. A spacer sequence, to aid in subsequent diagnostic PCRs was generated by polymerase cycling assembly (PCA). Briefly, the spacer sequence was synthesised by using primers of 60 bp length with 20 bp homologous sequence to the adjacent primers on each side. Final concentrations of 0.6 µM for outer primers (ol488 and ol492) and 0.03 µM of inner primers (ol489, ol490, ol491 and ol503 were used for PCA with the same cycle conditions as described for PCR. The final product was inserted with SpeI and NcoI restriction sites between the HRs as described for pDonor_ *pkdbpα* cloning, to replace the deleted DBPγ genes. Primer pairs are shown in *Figure 4—source data 2* andprimer sequences in *Figure 5—source data 2*.

## Three-step nested PCR

Generation of each PCR repair template was carried out by a three-step nested PCR method to fuse together HRs with the insert DNA (eGFP expression cassette, eGFP with N-terminal linker or mCherry with C-terminal linker). In a first set of PCRs, the DNA insert (eGFP expression cassette or tag) and the HRs for integration into the region of interest were individually amplified in duplicate. The HRs included at least 50 bp overhangs (OH) so that nested PCRs could be carried out in the next PCR step, without shortening the final size of HRs. Nested PCRs were carried out to increase the yield of PCR product. The HRs contained at least 20 bp and 58°C Tm overhangs with homology to the insert DNA (HR1 with C-term overhang homologous to the N-term of insert DNA and HR2 with N-term overhang homologous to the C-term of the insert DNA). All duplicates were pooled and products were extracted from agarose gel (Qiagen) to remove primers and background amplicons. In a second nested PCR HR1 was fused to the donor amplicon in duplicate with double the amount of time allowed for the elongation step (10 s/kb) and again the product was gel extracted. In the final step the HR1-insert and HR2 were fused together resulting in the final product HR1-insert-HR2 (*Figure 2A*). PCR repair templates for HA tagging were generated in a two-step PCR method. First the HRs were individually amplified with addition of 27 bp HA sequence overhangs on the 3'end of HR1 and the 5'end of HR2. In the second nested PCR HR1 and HR2 were fused.

All primers are listed in *Figure 5—source data 2* and all primer combinations for each contruct are listed in *Figure 2—source data 2*. Six to eight 50 µl reactions of the final construct PCRs were pooled (300 to 400 µl final volume and 20 µg DNA), ethanol precipitated and resuspended into sterile TE buffer for transfection. DNA concentrations were determined using Nano-Drop and band intensity measurement with BioRad Image lab software.

## DNA analysis

Genomic DNA from transfected parasite lines was extracted (QIAGEN) and analysed by PCR with GoTaq Master Mix (Promega) using the following conditions: 3 min at 96°C, then 30 cycles of 25 s at 96°C, 25 s at 52°C, and 1 min/kb at 64°C.

## Western blotting

To detect tagged proteins of interest, soluble cell extracts were prepared by lysing Nycodenz-enriched schizonts in 0.15% saponin. Parasite pellets were washed several times with cold PBS and centrifugation at 13,000 rpm for 3 min at 4°C to remove haemoglobin and red cell debris. Pellets were lysed in five pellet volumes of RIPA buffer (25 mM Tris, 150 mM NaCl, 1% Triton X-100, 0.5% Sodium deoxycholate, 0.1% SDS, pH 7.5, 1x Roche Protease Inhibitors) supplemented with 50 units BaseMuncher (Expedeon) on ice for 20 min. This whole cell lysate was clarified by centrifugation at 13,000 rpm for 30 min at 4°C. Soluble extracts were separated on Mini-Protean 4–20% TGX gels (Bio-Rad) and transferred to nitrocellulose using the Trans-blot Turbo system (Bio-Rad). Equivalent uninfected red cell lysate or wild type *P. knowlesi* schizont lysates were analysed alongside lysates

containing tagged proteins of interest. Membranes were blocked overnight, and tagged proteins were detected with mouse anti-GFP (Sigma, 1:5,000), rat anti-HA (Sigma 3F10 clone, 1:5,000), or rabbit anti-mCherry (Abcam, 1:5,000). Primary antibodies were detected using HRP-conjugated secondary antibodies (Bio-Rad, 1:5,000) and ECL (ThermoFisher Pierce). Chemiluminescence was captured using the Azure c600 system.

## Immunofluorescence assays and live cell imaging

Immunofluorescence assays were performed using blood smears fixed with 4% paraformaldehyde for 30 min followed by washing in PBS and permeabilisation in 0.1% Triton-X100 for 10 min. Slides of HA-tagged parasite lines were blocked overnight at 4°C in 3% bovine serum albumin/PBS and then labelled with rabbit anti-HA high affinity (1:250) and Alexa Fluor 488-conjugated α-rabbit IgG (1:5,000) (Thermo Fisher Scientific). The smears were mounted in ProLong Antifade mountant with DAPI (Thermo Fisher Scientific). For live cell imaging parasites were stained with Hoechst 33342 (New England Biolabs), transferred to poly-L-lysine-coated µ-slides VI (Ibidi, Martinsried, Germany). Both live and fixed preparations were viewed with a Nikon Ti E inverted microscope using a 100x oil immersion objective and imaged with an ORCA Flash 4.0 CMOS camera (Hamamatsu). Images were acquired and processed using the Nikon Elements Advanced Research software package.

## Parasite multiplication rate assays

In this flow cytometry-based assay the fold increase in parasitemia following one round of asexual growth is measured. Purified schizonts were set up in technical duplicate cultures with human RBCs, at a 2% hematocrit and ~0.5% parasitemia in 24 well plates. Parasitemia was measured with a flow cytometry (FACS)-based assay before and after incubation at 37°C in a gassed chamber for 24 hr. Samples were fixed with 2% paraformaldehyde (Sigma) and 0.2% glutaraldehyde (Sigma) in PBS for 1 hr at 4°C, washed, permeabilized with 0.3% Triton X-100, and then washed again before 1 hr RNase (MP Biomedicals) treatment, staining with SYBR Green I (Life Technologies), and FACS analysis. The samples were analyzed on a Becton Dickenson LSR-II. Data were acquired using FACSDiva 6.1.3 software and analyzed using FlowJo_V10.

Data was normalised to a 1% starting parasitemia to enable comparison of fold multiplication between lines. For comparison of growth in human and macaque blood, three biological independent experiments were carried out in *Macaca fascicularis* blood and eight biological independent experiments were carried out in human blood. Biological independent experiments include set up of experiments on different days with different parasite preparations and blood samples. Each biological replicate includes two technical replicates. For statistical analysis one-way ANOVA with Tukey's with multiple comparisons test of unpaired data was used. For investigation of the association between Duffy phenotype and growth rates, blood samples were collected from 21 volunteers, washed and then set up in technical dublicate cultures at a 2% haematrocrit and with ~0.5% parasitemia in 96 well plates. Parasites were maintained in 96-well microtiter plates for 24 hr and the parasitaemia monitored by FACS. Data of three independent schizont preparations were analysed for each blood sample.

## Blood typing

Venous blood (~10 mL) collected into EDTA Vacutainer (SLS) was anonymized, and then RBCs washed with RPMI 1640 and stored at 4°C. Blood Grouping Reagents for ABO, Rhesus D, and Duffy antigen (FyA, FyB) were used according to the manufacturer's instructions (Lorne Laboratories).

## Growth inhibition activity assays

Assays of growth inhibition activity (GIA), in the presence of anti-PvDBP_RII antibodies, were carried out using total IgG purified from rabbit sera using protein G columns (Pierce). Immunisation of rabbits against PvDBP_RII (Sall) has been described previously (*de Cassan et al., 2015*). Purified IgG was buffer-exchanged into RPMI 1640 medium, concentrated using ultra centrifugal devices (Millipore) and filter sterilized through a 0.22 µm filter (Millipore) prior to being aliquoted and frozen at −20°C until use.

*P. knowlesi* parasites were synchronized by magnetic separation (MACS LS columns, Miltenyi Biotech). Synchronized trophozoites were adjusted to 1.5% parasitemia, and 20 µL aliquots were

pipetted into 96-well flat/half area tissue culture cluster plates (Appleton Woods). 20 μL purified IgG were added to triplicate test wells at eight final concentrations (10, 5, 2.5, 1.25, 0.625, 0.312, 0.15 and 0.075 mg/mL) and incubated for one cycle (26–30 hr). Parasitemia was measured using the lactate dehydrogenase (pLDH) activity assay following standard protocols (*Kennedy et al., 2002*). An anti-DARC Fy6 VHH nanobody, a kind gift from Dr Olivier Bertrand (INSERM, France), was included in the test plate as a positive control in every assay (final concentration 1.5 or 3 μg/mL) and purified control IgG from the pre-immunisation sera of matched rabbits were used as the negative control. Anti-PkMSP119 rabbit sera, a kind gift from Ellen Knuepfer (Crick Institute, UK), was also tested in a similar manner. GIA of the purified IgG was expressed as percent inhibition calculated as follows: 100 − [(OD650 of infected erythrocytes with test IgG − OD650 of normal erythrocytes only) / (OD650 of infected erythrocytes without any IgG − OD650 of normal erythrocytes only) x 100%].

## Whole genome sequencing

Genomic DNA was prepared for the PvDBP$^{OR}$/Δ14Δγ using the DNeasy Blood and Tissue Kit (Qiagen). DNA libraries were prepared using the QIAseq FX DNA Library Kit (Qiagen) as per manufacturer's instructions. A 20 min fragmentation step was optimized for *Plasmodium* samples. Whole genome sequencing was performed using Illumina MiSeq technology with 150-base paired end fragment sizes. Raw sequence data for the A1-H.1 parental line was extracted from the European Nucleotide Archive as per (*Moon et al., 2016*; *Benavente et al., 2018*). The raw sequence data (accession number ERS3042513) was processed as previously described (*Campino et al., 2016*). In brief, the raw sequence data was aligned onto the A1-H.1 reference genome using the *bwa-mem* short read alignment algorithm, and coverage statistics were obtained using the *sambamba* software to be plotted using R.

## Acknowledgements

This work is supported by an MRC Career Development Award (MR/M021157/1) jointly funded by the UK Medical Research Council and Department for International Development (RWM, FM). MNH is supported by a Bloomsbury Colleges research studentship. JAC is supported by a BBSRC LIDO studentship. TAR held a Wellcome Trust Research Training Fellowship [108734/Z/15/Z]; TGC is funded by the Medical Research Council UK [ref. MR/M01360X/1, MR/N010469/1, MR/R025576/1, and MR/R020973/1] and BBSRC [ref. BB/R013063/1]. SC is funded by Medical Research Council UK grants [ref. MR/M01360X/1, MR/R025576/1, and MR/R020973/1]. SJD is a Jenner Investigator, a Lister Institute Research Prize Fellow and a Wellcome Trust Senior Fellow [106917/Z/15/Z]. We are grateful to the Wellcome Trust for a Senior Investigator Award to DAB (ref. 106240/Z/14/Z). RCH is supported by a Marshall Scholarship granted by Her Majesty's Government. CJS is supported by Public Health England. We thank Jose-Juan Lopez-Rubio for providing the pUF and pl6 plasmid, and also David Llewellyn, Jennifer Marshall and Doris Quinkert (University of Oxford) for assistance with rabbit immunisations, parasite culturing and the assays of GIA. We thank Michael Blackman (Francis Crick Institute) for providing Compound 2. We also thank Carolynne Stanley for recruiting volunteers and drawing of blood samples.

# Additional information

## Funding

| Funder | Grant reference number | Author |
|---|---|---|
| Medical Research Council | MRC Career Development Award MR/M021157/1) | Franziska Mohring Robert William Moon |
| University of London | Bloomsbury Colleges Research Studentship | Melissa Natalie Hart |
| Wellcome Trust | Research Training Fellowship, 108734/Z/15/Z | Thomas A Rawlinson |
| Marshall Aid Commemoration Commission | Marshall Scholarship | Ryan Henrici |

| Biotechnology and Biological Sciences Research Council | London Interdiscpinary Doctoral Programme Studentship | James A Charleston |
|---|---|---|
| Wellcome Trust | Senior Investigator Award, 106240/Z/14/Z | Avnish Patel<br>David A Baker |
| Medical Research Council | MR/M01360X/1 | Susana Campino<br>Taane G Clark |
| Biotechnology and Biological Sciences Research Council | BB/R013063/1 | Taane G Clark |
| Medical Research Council | MR/R025576/1 | Taane G Clark |
| Medical Research Council | MR/R020973/1 | Taane G Clark |
| Medical Research Council | MR/N010469/1 | Taane G Clark |
| Public Health England | | Colin J Sutherland |
| Lister Institute of Preventive Medicine | Research Prize Fellow | Simon J Draper |
| Wellcome Trust | Wellcome Trust Senior Fellowship, 106917/Z/15/Z | Simon J Draper |

The funders had no role in study design, data collection and interpretation, or the decision to submit the work for publication.

### Author contributions

Franziska Mohring, Conceptualization, Formal analysis, Validation, Investigation, Visualization, Methodology, Writing—original draft, Writing—review and editing; Melissa Natalie Hart, Data curation, Formal analysis, Validation, Investigation, Visualization, Methodology, Writing—review and editing; Thomas A Rawlinson, Formal analysis, Validation, Investigation, Visualization, Methodology, Writing—original draft, Writing—review and editing; Ryan Henrici, Investigation, Visualization, Methodology, Writing—original draft, Writing—review and editing; James A Charleston, Data curation, Formal analysis, Validation, Investigation, Methodology, Writing—review and editing; Ernest Diez Benavente, Data curation, Formal analysis, Investigation, Methodology, Writing—review and editing; Avnish Patel, Resources, Investigation, Methodology, Writing—review and editing; Joanna Hall, Resources, Investigation, Writing—review and editing; Neil Almond, Resources, Supervision, Writing—review and editing; Susana Campino, Resources, Supervision, Funding acquisition, Investigation, Writing—review and editing; Taane G Clark, Colin J Sutherland, Supervision, Funding acquisition, Writing—review and editing; David A Baker, Resources, Supervision, Funding acquisition, Writing—review and editing; Simon J Draper, Conceptualization, Resources, Supervision, Funding acquisition, Methodology, Project administration, Writing—review and editing; Robert William Moon, Conceptualization, Resources, Data curation, Formal analysis, Supervision, Funding acquisition, Investigation, Visualization, Methodology, Writing—original draft, Project administration, Writing—review and editing

### Author ORCIDs

Franziska Mohring https://orcid.org/0000-0003-4726-6693
Taane G Clark http://orcid.org/0000-0001-8985-9265
Colin J Sutherland http://orcid.org/0000-0003-1592-6407
Simon J Draper http://orcid.org/0000-0002-9415-1357
Robert William Moon https://orcid.org/0000-0002-9070-4292

### Ethics

Human subjects: For most experiments and routine parasite maintenance, human blood (Duffy (FY) positive) was obtained from the United Kingdom National Blood Transfusion Service under a research agreement. For the Duffy subtype experiment, venous blood (~10 mL) from 21 healthy volunteers blood donors was collected after donors had provided informed consent. Blood samples were then anonymized. The project, consent and protocol were approved by the LSHTM Observational Research Ethics Committee under project reference 5520-1.

Animal experimentation: Animal work was reviewed and approved by the local National Institute for Biological Standards and Control Animal Welfare and Ethical Review Body (the Institutional Review Board) and by the United Kingdom Home Office as governed by United Kingdom law under the Animals (Scientific Procedures) Act 1986. Animals were handled in strict accordance with the "Code of Practice Part 1 for the housing and care of animals (21/03/05)" available at https://www.gov.uk/research-and-testing-using-animals. The work also met the National Centre for the Replacement Refinement and Reduction of Animals in Research (NC3Rs) guidelines on primate accommodation, care, and use (https://www.nc3rs.org.uk/non-human-primate-accommodation-care-and-use), which exceed the legal minimum standards required by the United Kingdom Animals (Scientific Procedures) Act 1986, associated Codes of Practice, and the US Institute for Laboratory Animal Research Guide.

### Decision letter and Author response
Decision letter https://doi.org/10.7554/eLife.45829.027
Author response https://doi.org/10.7554/eLife.45829.028

## Additional files

### Supplementary files
• Transparent reporting form
DOI: https://doi.org/10.7554/eLife.45829.023

### Data availability
Raw whole genome sequencing data has been deposited to the European Nucleotide Archive under accession number ERS3042513. All other data generated or analysed during this study are included in the manuscript and supporting files. Source data files for all figures has been provided.

The following dataset was generated:

| Author(s) | Year | Dataset title | Dataset URL | Database and Identifier |
|---|---|---|---|---|
| Mohring F, Hart MN, Rawlinson TA, Henrici R, Charleston JA, Benavente ED, Patel A, Hall J, Almond N, Campino S, Clark TG, Sutherland CJ, Baker DA, Draper SJ, Moo RW | 2019 | Genome editing in the zoonotic malaria parasite Plasmodium knowlesi provides new tools for P. vivax research | http://www.ebi.ac.uk/ena/data/view/ERS3042513 | European Nucleotide Archive, ERS3042513 |

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
