## [Decision Letter]

Thank you for submitting your article "Rapid and iterative genome editing in the zoonotic malaria parasite *Plasmodium knowlesi:* New tools for *P. vivax* research" for consideration by *eLife*. Your article has been reviewed by Gisela Storz as the Senior Editor, a Reviewing Editor, and three reviewers. The following individuals involved in review of your submission have agreed to reveal their identity: Tobias Spielmann (Reviewer #1).

The reviewers have discussed the reviews with one another and the Reviewing Editor has drafted this decision to help you prepare a revised submission.

Summary:

Mohring et al., present a system to edit the genome of P. knowlesi, the source of zoonotic malaria and model for the difficult to study *P. vivax*. They use CRISPR/CAS to tag or knock out genes in a manner that the parasites obtained are marker free, permitting multiple serial modifications. This is achieved using positive and negative selection cassettes on the plasmid carrying the CRISPR/Cas9 components to generate the genomic break together with a marker free PCR product (or plasmid) for repair. This study is major advance of *Plasmodium knowlesi* as a laboratory model.

The versatility of this system is validated by a relatively thorough functional analysis of the DBP orthologs of *P. knowlesi* and *P. vivax* and the DBP-like erythrocyte binding proteins (β, γ). This exemplary dissection of host cell specificity of the parasite ligands provides both important confirmations and corrections of their previously-defined functions. Finally, the study uses a *P. knowlesi* line completely dependent on PvDBP to evaluate efficacy of anti-PvDBP IgG. Demonstrating the utility of the PvDBP-transgenic *P. knowlesi* for this purpose is an important breakthrough for *vivax* malaria research to develop asexual blood-stage vaccines and other targeted therapeutics against *P. vivax*.

Essential revisions:

1) DBPs/EBPs have clearly defined roles as invasion ligands. Consequently, it is not optimal to use multiplication rate as the functional assay to assess differences in invasion. The Results section and Discussion section further erroneously conflate growth with invasion as well as with tropism. At a minimum the terminology should be appropriate with the analytical methods. Multivariate statistical analyses would be appropriate to analyze these data. Supplemental tables with the growth/invasion and GIA data in numerical format are needed.

2) The description of the 44 kb deletion including PkDBP-β is a bit confusing. Please revise the sentences in subsection “Transgenic *P. knowlesi* orthologue replacement lines provide surrogates for *P. vivax* vaccine

development and DBP tropism studies”. It appears as the parental line is a mixture of parasites with and without the deletion. Clones containing PkDBP-β (i.e., without the deletion) were not further analyzed. Thus, in Figure 4B all transgenic lines are deficient for PkDBP-β, and the WT line is a mixture of parasites with or without this gene. This precludes conclusions about the role of PkDBP-β. Suggestion: (i) determine the proportion of parasites in the WT population that have PkDBP-β, e.g. by Southern blot; (ii) include clones positive for PkDBP-β in the analysis presented in Figure 4B; (iii) otherwise, review the statements about the role of this gene throughout the paper.

3) In addition to the difference described by the authors, which focuses on the PvDBPa-OR-δ-β-γ line, there appears to be an increase of a similar magnitude in mRBCs invasion by PkDBP-OR lines compared to the WT line. This increase is not mentioned but it may have a biological meaning: deletion of DBP β and γ may result in use of different, more efficient invasion pathways. The increase observed should be discussed. The statistical analysis should preferably be presented in the figure, rather than in the legend, and all significant differences indicated.

4) The final paragraph of the Discussion section does not accurately reflect publications in this area of study. Whilst the *P. knowlesi* OR lines do represent an important breakthrough, much of what is stated to be achievable with the transgenic *P. knowlesi* has been learned already using various in vitro DBP functional assays expressing the DBP ligand domain. Indeed, these recombinant proteins assays are a vital attribute to the field and have been used to map epitopes, characterize antigenically-distinct DBP allelic variants, and define functional motifs of DBP. These findings have been supported by structural studies and the primary structural model cited indicates the protective effect of immune inhibitory antibodies is to prevent dimerization of PvDBP instead of directly block receptor engagement.

---

## [Author Response]

Essential revisions:1) DBPs/EBPs have clearly defined roles as invasion ligands. Consequently, it is not optimal to use multiplication rate as the functional assay to assess differences in invasion. The Results section and Discussion section further erroneously conflate growth with invasion as well as with tropism. At a minimum the terminology should be appropriate with the analytical methods. Multivariate statistical analyses would be appropriate to analyze these data. Supplemental tables with the growth/invasion and GIA data in numerical format are needed.

The reviewers raise an important point here. The multiplication rate assays use purified synchronised schizonts added to test cells and allowed to invade and grow for 24 hours followed by direct measurement of parasite numbers by FACs. We know from previous work (1, 2), that unless there is an additional phenotype during intracellular growth the parasitaemia after 3-6 hours post invasion is the same as that seen at 24 hours. Given the clearly defined role of DBPs in invasion, we therefore opted to use this approach, which enabled us to carry out many more independent repeats than a more targeted invasion assay would have allowed. Whilst we are working on more specific invasion assays, shorter invasion windows can be very challenging given the propensity of *P. knowlesi* merozoites to remain invasive over relatively long periods (3), as well as difficulties achieving extremely tight synchrony in 5 different lines simultaneously. Whilst we looked into carrying out additional targeted invasion assays in response, we were unfortunately unable to secure sufficient donations of macaque blood within the allotted timeframe to do so. We completely agree with the reviewers, that conflation of growth and invasion is erroneous and so have further refined the language used to describe the assays and findings. We therefore included the definition in the methods sections and changed the terminology from invasion rate to multiplication rate in the manuscript where appropriate.

Summary of text changes:

Subsection “Transgenic *P. knowlesi* orthologue replacement lines provide surrogates for *P. vivax* vaccine development and DBP tropism studies”: “It also demonstrated, that not only are PkDBPβ and γ proteins dispensable for multiplication in humans RBCs (4), but as their loss leads to increased multiplication rate, they may actually impede invasion in human RBCs. A further experiment comparing multiplication rates in 21 Duffy positive blood donations revealed significant higher growth rates of the PkDBPα^OR^/Δ14Δγ….”

“Interestingly, loss of DBPβ and DBPγ in the PkDBPα^OR^/Δ14Δγ line did not reduce the parasite multiplication rate in macaque RBC….”

“Whilst macaque multiplication rates and host preference ratio….”

“40% reduction in macaque multiplication rates (Figure 4B) which caused a shift….”

Discussion section: “Nevertheless as the multiplication rate remained quite close….”

Subsection “Parasite multiplication rate assays”

Figure 3 legend: “Bar chart showing mean fold replication of parasites lines in FACS-based multiplication assays over one growth cycle….”

Further we have changed the term “host cell tropism” of *P. knowlesi* to human and macaque RBCs to “host cell preference”.

Subsection “Transgenic *P. knowlesi* orthologue replacement lines provide surrogates for *P. vivax* vaccine development and DBP tropism studies”: “To investigate how these modifications affect host preference we….”

Discussion section: “By applying these techniques to the *P. knowlesi* and *P. vivax* DBP family we have been able to examine the role of these genes in host and reticulocyte preference of the two species.”

Multiplication and GIA assays have been divided into Figure 4 and Figure 5 and datasets have been included as figure supplements.

The graph in Figure 4C compares the growth of 5 parasite lines in two different host cells (human and macaque) and the reviewers suggest using multivariate statistical analyses. While we thank the authors for this suggestion, as only one dependent variable is measured (fold multiplication) we decided to use a one-way ANOVA instead. We have changed the layout of the figure by grouping the data for human or macaque blood to make it easier to highlight all statistical comparisons.

2) The description of the 44 kb deletion including PkDBP-β is a bit confusing. Please revise the sentences in subsection “Transgenic P. knowlesi orthologue replacement lines provide surrogates for P. vivax vaccine development and DBP tropism studies”. It appears as the parental line is a mixture of parasites with and without the deletion. Clones containing PkDBP-β (i.e., without the deletion) were not further analyzed. Thus, in Figure 4B all transgenic lines are deficient for PkDBP-β, and the WT line is a mixture of parasites with or without this gene. This precludes conclusions about the role of PkDBP-β. Suggestion: (i) determine the proportion of parasites in the WT population that have PkDBP-β, e.g. by Southern blot; (ii) include clones positive for PkDBP-β in the analysis presented in Figure 4B; (iii) otherwise, review the statements about the role of this gene throughout the paper.

This was a slightly confusing situation and we agree that we could improve our explanation. Whilst originally clonal, the A1-H.1 line used for transfections to replace PkDBPα with PkDBPα^OR^/PvDBP^OR^ had been grown for several months prior to transfection during which time a 44kb truncation in Chromosome 14 must have occurred. We didn’t realise this until sometime after transfection and stocks of this particular Wild Type were not retained, but clones from other transfections with the same line confirmed that the chromosome 14 truncation (now termed Δ14) must have been the dominant clone. Whole genome sequencing then confirmed the deletion. The *P. knowlesi* lines are very recently adapted to culture and as such probably less “stable” than lines such as Pf3D7. It seems likely that the truncation offered the parasites a small growth advantage and thus outgrew the original parasite population without truncation. Unfortunately, we don’t have stocks of this wtΔ14 population anymore to undertake further investigation, but all multiplication assays used a true WT line that retained an intact chromosome 14. The nature of the change would also make it difficult to give a proportion of parasites with DBPβ loss within a wild type population by PCR. We repeated this again using a different wild type stock without truncation to generate an independent second PkDBPα^OR^ clonal line to generate a CRISPR based DBPβ knockout. We continued using the Δ14 lines for subsequent work as we reasoned that there would be a danger that it may end up happening again anyway.

We have rewritten this paragraph to make it clearer for the reader. We are renaming the WT line with truncation to WtΔ14, and PkDBPα^OR^/Δβ into PkDBPα^OR^/Δ14. We include diagnostic PCRs of the CRISPR-generated a PkDBPβ knockout line PkDBPα^OR^/Δβ. The CRISPR scheme for DBPβ k.o. and alignment of the PkDBPβ guide has been included. The scheme of generated transgenic lines has been changed accordingly.

We include a further experiment comparing the following lines in multiplication rate assays; Wild type, PkDBPα^OR^ (without truncation), and PkDBPα^OR^/Δβ. Here, we show that neither PkDBPα recodonisation nor PkDBPβ knockout affects multiplication rates in human RBCs.

Changes in the manuscript subsection “Transgenic *P. knowlesi* orthologue replacement lines provide surrogates for *P. vivax* vaccine development and DBP tropism studies”:

“Whole genome sequencing and mapping against the A1-H.1 reference genome revealed that the PkDBPα^OR^ and PvDBP^OR^ line have a ~44 kb truncation at one end of chromosome 14 (Figure 4—figure supplement 2), which also harbours DBPβ, therefore we renamed the lines PkDBPα^OR^/Δ14 and PvDBP^OR^/Δ14.”

“To confirm this, another PkDBPα^OR^ clonal line was generated in an independent transfection using the A1-H.1 parental parasite line with intact DBPβ locus.”

“Reasoning that as the Δ14 truncation may have provided a selective advantage to the parasites and thus may well reoccur, we took advantage of the spontaneous DBPβ loss (PkDBPα^OR^/Δ14 or PvDBP^OR^/Δ14) and then used pCas9/sg plasmid recycling to additionally delete the DBPγ locus (Figure 4—figure supplement 1B), generating PkDBPα^OR^/Δ14Δγ and PvDBP^OR^/Δ14Δγ. An overview of all generated DBP lines is depicted in Figure 4—figure supplement 1E).”

“Using the PkDBPα^OR^ clonal line with the start of chromosome 14 intact, we were able to generate a Cas9 mediated DBPβ knockout (PkDBPα^OR^/Δβ) (Figure 4B and Figure 4—figure supplement 1C). FACS-based multiplication assays showed no growth effects of PkDBPβ knockout in human blood (Figure 4—figure supplement 1F).”

3) In addition to the difference described by the authors, which focuses on the PvDBPa-OR-δ-β-γ line, there appears to be an increase of a similar magnitude in mRBCs invasion by PkDBP-OR lines compared to the WT line. This increase is not mentioned but it may have a biological meaning: deletion of DBP β and γ may result in use of different, more efficient invasion pathways. The increase observed should be discussed. The statistical analysis should preferably be presented in the figure, rather than in the legend, and all significant differences indicated.

In response to this and comment 1 we have altered our statistical analysis and now use a one-way ANOVA with Tukey’s test for multiple comparisons and compare all tested lines with each other. Both this and the previous t-tests are adjusted for multiple comparisons and this decreases the statistical power. As such, despite fairly large differences in the multiplication rates the only significant comparisons are the WT/ PkDBPα^OR^/Δ14Δγ line in human blood and the PkDBPα^OR^/Δ14Δγ and PvDBP^OR^/Δ14Δγ lines in macaque blood. In human blood we carried out 8 biological independent experiments (different blood, different parasite prep for each) with two technical replicates per experiment. Due to limited available macaque blood, we could only carry out three biological independent experiments with two technical replicates per experiments. With analysis of more biological replicates we might have been able to see additional significant changes in growth rates due to the deletion of the paralogues. However, we agree that the trends in the available data indicate that PkDBPα is the preferred pathway for invasion in both human and macaque cells and will highlight this in the discussion (with relevant caveats).

The Discussion section was changed to:

“Deletion of these genes led to a significant increase in growth rates in human RBCs, and interestingly also a similar, but non-significant, increase in growth rates in macaque RBCs. This suggests that PkDBPα alone is sufficient to retain the multiplication capacity and that the PkDBP paralogues are not responsible for the macaque cell preference retained in the A1-H.1 human adapted line. Whilst likely non-functional for invasion of human RBCs, both PkDBPγ and DBPβ would still compete for surface space and potentially some of the same interacting proteins as DBPα, which may account for the increased growth rate when they are deleted. The increased growth rate in macaque cells, would suggest that even here, where DBPγ and DBPβ are functional, PkDBPα is the preferred and more efficient pathway for invasion in macaque cells. In vivo, this advantage would be countered by the significant benefits provided by redundancy both to combat host blood cell polymorphisms and antibody responses. Importantly, this data suggests that loss of DBPγ and DBPβ could provide an adaptation route to increased growth/virulence within human infections. However, to do so the parasites would need to sacrifice redundancy within its primary macaque hosts – a situation only likely to occur with prolonged human to human transmission. Further work to analyse the effect of these mutations through long-term parasite competition assays will enable us to determine the precise extent of this advantage and determine how quickly such mutations could move to fixation within a parasite population.”

We have included another experiment comparing growth rates of the three lines wt, PkDBPα^OR^/Δ14Δγ and PvDBP^OR^/Δ14Δγ in Duffy positive RBCs from 21 individuals. The increased number of bloods increased statistical power and we could further confirm that the growth rates of PkDBPα^OR^/Δ14Δγ line are significantly enhanced. This also demonstrates that the PvDBP^OR^/Δ14Δγ.

We included the following sentences to the Results section:

“Analysis of the wild type line, and the four transgenic lines (PkDBPα^OR^/Δ14, PvDBP^OR^/Δ14, PkDBPα^OR^/Δ14Δγ and PvDBP^OR^/Δ14Δγ) revealed no difference in growth rate in human RBCs for all lines except the PkDBPα^OR^/Δ14Δγ which demonstrated significantly increased growth rate compared to wild type (Figure 4C and Figure 4—figure supplement 1F). This confirmed that the *P. vivax* protein was able to complement the role of its *P. knowlesi* orthologue. It also demonstrated, that not only are PkDBPβ and γ proteins dispensable for multiplication in human RBCs (4), but as their loss leads to increased multiplication rate, they may actually impede invasion in human RBCs. A further experiment comparing multiplication rates in 21 Duffy positive blood donations revealed significant higher growth rates of the PkDBPα^OR^/Δ14Δγ line (8.9 fold) compared to the 6.9 fold multiplication of the wild type line and PvDBP^OR^/Δ14Δγ line (p< 0.001) (Figure 4D and Figure 4—figure supplement 4C and D), confirming that knockout of the two paralogues increases multiplication rate in human RBCs but also revealing a minor growth reduction in the PvDBP^OR^/Δ14Δγ line compared to its control line.”

4) The final paragraph of the Discussion section does not accurately reflect publications in this area of study. Whilst the P. knowlesi OR lines do represent an important breakthrough, much of what is stated to be achievable with the transgenic P. knowlesi has been learned already using various in vitro DBP functional assays expressing the DBP ligand domain. Indeed, these recombinant proteins assays are a vital attribute to the field and have been used to map epitopes, characterize antigenically-distinct DBP allelic variants, and define functional motifs of DBP. These findings have been supported by structural studies and the primary structural model cited indicates the protective effect of immune inhibitory antibodies is to prevent dimerization of PvDBP instead of directly block receptor engagement.

We agree that this section is unnecessarily brief and did not do justice to the many studies that underpin our current understanding of DBP/DARC interactions. Our intention was not to undermine the value of this work in any way. However, we do think that binding assays and structural studies using recombinant protein alone are not sufficient to fully understand the functional role of DBP or mechanisms of inhibition and feel our lines offer the opportunity for a range of new insights. This view is supported by the fact that many of the labs involved in these prior studies have already been in touch to request access to our lines (which we will of course provide). We already highlight that the transgenic lines offer the potential to enable functional analysis of full-length PvDBP, rather than just recombinant fragments of region II, as well as offering the chance to study DBP function in its endogenous context at the interface between the RBC and parasite surface – where, for example, epitopes may be concealed by other proteins. We explore some of these differences in more detail in a follow-on paper which was submitted to the editorial team as an associated paper to this work. This has recently been accepted in Nature Microbiology (5).

In view of this we have added some additional background to highlight the valuable role that recombinant work has served to date, as well as highlighting the benefits of an integrative approach combining a variety of methods.

The following was added to the Discussion section:

“Using a combination of binding assays, mutagenesis and structural studies with recombinant PvDBP-RII, previous work has identified residues involved in DBP-DARC interactions (6-8), as well as a DBP dimerization domain thought to drive the stepwise engagement with DARC (9, 10). Efforts to develop a *P. vivax* vaccine to elicit antibodies against the lead candidate PvDBP have predominantly relied on using ELISA-based assays, which assess the ability of antibodies to block recombinant PvDBP-RII binding to DARC (11). This has successfully identified a range of binding inhibitory antibodies mapped to residues involved in DARC binding, dimerization as well as subdomain 3, which is distant from the DARC binding site, demonstrating that multiple mechanisms of inhibition may be important (12, 13). However, validation of these findings in a parasitological assay is critical. Some epitopes identified in recombinant protein assays may be inaccessible in the context of invasion and it is also possible that not all inhibitory antibodies block receptor binding.”

“Whilst the relationship between in vitro GIA and in vivo protection against malaria challenge has not always been clear, previous studies have shown that there is a strong correlative link between activity in the in vitro assay of GIA and in vivo protection from challenge of two different species of human malaria (*P. falciparum* and *P. knowlesi*) in two different non-human primate models (Aotus monkeys and rhesus macaques) targeting three different merozoite proteins (PfAMA1, PfRH5, PfMSP1) (14-16). Importantly, Douglas *et al.* in 2015 determined a threshold value for in vitro GIA assay which must be reached before protection from malaria challenge is afforded (16). Thus, the ability to readily assess *P. vivax* candidates in an in vitro GIA offers an important mechanism to determine whether candidates are appropriate for moving on to challenge studies.”

“Here we use the PvDBP sequence from the SalI reference strain, but multiple lines expressing distinct PvDBP variants could be generated in future, as has already been used for recombinant ELISA based assays (17), to systematically examine inhibition in heterologous strains.”

1. Moon RW, Hall J, Rangkuti F, Ho YS, Almond N, Mitchell GH, et al. Adaptation of the genetically tractable malaria pathogen *Plasmodium knowlesi* to continuous culture in human erythrocytes. Proc Natl Acad Sci U S A. 2013;110(2):531-6.

2. Moon RW, Sharaf H, Hastings CH, Ho YS, Nair MB, Rchiad Z, et al. Normocyte-binding protein required for human erythrocyte invasion by the zoonotic malaria parasite *Plasmodium knowlesi*. Proc Natl Acad Sci U S A. 2016;113(26):7231-6.

3. Lyth O, Vizcay-Barrena G, Wright KE, Haase S, Mohring F, Nejer A, et al. Cellular dissection of malaria parasite invasion of human erythrocytes using viable Plasmodium knowlesi merozoites. Scientific Reports. 2018;8.

4. Dankwa S, Lim C, Bei AK, Jiang RHY, Abshire JR, Patel SD, et al. Ancient human sialic acid variant restricts an emerging zoonotic malaria parasite. Nature Communications. 2016;7.

5. Rawlinson TA, Barber NM, Mohring F, Cho JS, Kosaisavee V, Gerard SF, et al. Structural basis for inhibition of Plasmodium vivax invasion by a broadly neutralizing vaccine-induced human antibody. Nature microbiology. 2019.

6. Choe H, Moore MJ, Owens CM, Wright PL, Vasilieva N, Li W, et al. Sulphated tyrosines mediate association of chemokines and Plasmodium vivax Duffy binding protein with the Duffy antigen/receptor for chemokines (DARC). Molecular Microbiology. 2005;55(5):1413-22.

7. Hans D, Pattnaik P, Bhattacharyya A, Shakri AR, Yazdani SS, Sharma M, et al. Mapping binding residues in the Plasmodium vivax domain that binds Duffy antigen during red cell invasion. Molecular Microbiology. 2005;55(5):1423-34.

8. VanBuskirk KM, Sevova E, Adams JH. Conserved residues in the Plasmodium vivax Duffy-binding protein ligand domain are critical for erythrocyte receptor recognition. Proceedings of the National Academy of Sciences of the United States of America. 2004;101(44):15754-9.

9. Batchelor JD, Malpede BM, Omattage NS, DeKoster GT, Henzler-Wildman KA, Tolia NH. Red blood cell invasion by Plasmodium vivax: structural basis for DBP engagement of DARC. PLoS Pathog. 2014;10(1):e1003869.

10. Batchelor JD, Zahm JA, Tolia NH. Dimerization of Plasmodium vivax DBP is induced upon receptor binding and drives recognition of DARC. Nature Structural & Molecular Biology. 2011;18(8):908-U67.

11. Shakri AR, Rizvi MMA, Chitnis CE. Development of Quantitative Receptor-Ligand Binding Assay for use as a tool to estimate Immune Responses against *Plasmodium vivax* Duffy Binding Protein Region II. Journal of Immunoassay & Immunochemistry. 2012;33(4):403-13.

12. Chen E, Salinas ND, Huang YN, Ntumngia F, Plasencia MD, Gross ML, et al. Broadly neutralizing epitopes in the Plasmodium vivax vaccine candidate Duffy Binding Protein. Proceedings of the National Academy of Sciences of the United States of America. 2016;113(22):6277-82.

13. Chootong P, Ntumngia FB, VanBuskirk KM, Xainli J, Cole-Tobian JL, Campbell CO, et al. Mapping epitopes of the Plasmodium vivax Duffy binding protein with naturally acquired inhibitory antibodies. Infect Immun. 2010;78(3):1089-95.

14. Singh S, Miura K, Zhou H, Muratova O, Keegan B, Miles A, et al. Immunity to recombinant Plasmodium falciparum merozoite surface protein 1 (MSP1): Protection in Aotus nancymai monkeys strongly correlates with anti-MSP1 antibody titer and in vitro parasite-inhibitory activity. Infection and Immunity. 2006;74(8):4573-80.

15. Hamid MMA, Remarque EJ, van Duivenvoorde LM, van der Werff N, Walraven V, Faber BW, et al. Vaccination with Plasmodium knowlesi AMA1 Formulated in the Novel Adjuvant Co-Vaccine HT (TM) Protects against Blood-Stage Challenge in Rhesus Macaques. Plos One. 2011;6(5).

16. Douglas AD, Baldeviano GC, Lucas CM, Lugo-Roman LA, Crosnier C, Bartholdson SJ, et al. A PfRH5-Based Vaccine Is Efficacious against Heterologous Strain Blood-Stage Plasmodium falciparum Infection in Aotus Monkeys. Cell Host & Microbe. 2015;17(1):130-9.

17. Ntumngia FB, Schloegel J, Barnes SJ, McHenry AM, Singh S, King CL, et al. Conserved and Variant Epitopes of Plasmodium vivax Duffy Binding Protein as Targets of Inhibitory Monoclonal Antibodies. Infection and Immunity. 2012;80(3):1203-8.